# Detecting and Perturbing Privacy-Sensitive Neurons to Defend Embedding Inversion Attacks

## Abstract

This paper introduces Defense through Perturbing Privacy Neurons (DPPN), a novel approach to protect text embeddings against inversion attacks. Unlike existing methods that add noise to all embedding dimensions for general protection, DPPN identifies and perturbs only a small portion of privacy-sensitive neurons. We present a differentiable neuron mask learning framework to detect these neurons and a neuron-suppressing perturbation function for targeted noise injection. Experiments across six datasets show DPPN achieves superior privacy-utility trade-offs. Compared to baseline methods, DPPN reduces more privacy leakage by 5-78% while improving downstream task performance by 14-40%. Tests on real-world sensitive datasets demonstrate DPPN's effectiveness in mitigating sensitive information leakage to 17%, while baseline methods reduce it only to 43%.

## 1 Introduction

Text embeddings are general representations of textual data, allowing users to conduct various downstream learning without having to access or reveal the raw text data. Advancements in pre-trained models like Sentence-T5 Ni et al. (2022a) and Sentence-BERT Reimers & Gurevych (2019) allow users to leverage these models for generating high-quality embeddings. These embeddings power a wide range of NLP applications. Retrieval-augmented generation (RAG) systems Lewis et al. (2020) are a prime example that has fueled the adoption of online embedding database services like Chroma[1] and Faiss Johnson et al. (2019). In these databases, only the text embeddings are shared with third-party services, not the actual text. Since only encoded data (i.e., embeddings) is shared, there is a common misconception that privacy is well-preserved through this mechanism. Nevertheless, research has shown that attackers can infer sensitive information by conducting embedding inversion attacks with a reasonable success rate Li et al. (2023); Pan et al. (2020); Song & Raghunathan (2020). Recent work Vec2text Morris et al. (2023) further reveals that an adversary can recover 92% of a 32-token text input given embeddings from a T5-based pre-trained transformer. Such vulnerabilities are particularly concerning in scenarios where sensitive information like medical records or financial data is embedded.

To defend against embedding inversion attacks, perturbing text embeddings by injecting random noises is a widely used approach. For instance, previous works Pan et al. (2020); Morris et al. (2023) often add Laplace noises in text embeddings to counteract embedding inversion attacks. Existing noisy embedding methods often add random noise to all embedding dimensions for general protection. The drawbacks of this approach are twofold. First, although adding noise to all dimensions can protect sensitive information, it can also alter non-sensitive information embedded within the text, thereby degrading the performance of downstream tasks. Second, adding random noise uniformly across all dimensions might not be ideal, as some parts of the embeddings might require larger perturbations while others do not. Therefore, this work aims to address a key research question:

**Research Question:** *Is it possible to manipulate as few embedding dimensions as possible to protect sensitive information while minimizing perturbation to the non-sensitive parts?*

---

[1] https://docs.trychroma.com/

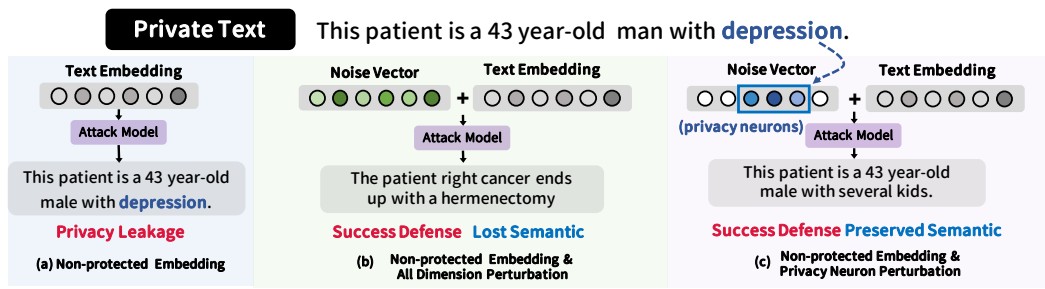

Figure 1: Illustration of the privacy-utility tradeoff on text embeddings. (a): The sensitive information could be easily identified with non-protected text embedding. (b): Perturbing text embedding on all dimensions prevents privacy leakage but damages the textual semantics. (c): Our DPPN perturbs text embedding selectively on privacy neurons, which protects privacy while maintaining non-sensitive textual semantics.

In essence, we aim to identify a set of so-called privacy neurons (i.e., embedding dimensions) within the representations that correlate with the private information to be protected. Take Figure 1 as an example: (a) demonstrates an example of privacy leakage caused by an embedding inversion attack. (b) represents previous models that add random noise to all dimensions. If the noise is large enough, it is possible to protect the sensitive information (i.e., depression in this example) at the cost of seriously altering the original meaning of the data. (c) illustrates our solution that identifies the privacy neurons associated with the term *depression* and selectively perturbs these dimensions to obfuscate the attack model. The ultimate goal is to not only protect the sensitive information but also ensure the non-sensitive information is still correctly encoded in the embeddings.

To achieve the above goal, this work focuses on addressing two follow-up research questions. First, *how to identify a subset of neurons associated with a given sensitive concept*; and second, *after identifying such neurons, how to manipulate their values to defend the attack*. In this work, we present the **D**efense through **P**erturbing **P**rivacy **N**eurons (DPPN) framework for a better privacy-preserving text embedding. Specifically, we first leverage a differentiable neuron mask learning framework to identify the top-$k$ privacy neurons associated with a target token $t$ to be protected. Given the detected neurons, we introduce a neuron-suppressing perturbation function to obfuscate the privacy information through directional noise injection. To fully evaluate the effectiveness of DPPN, we conduct comprehensive experiments and summarize the findings as follows:

- **Better privacy-utility tradeoffs.** We evaluated DPPN on six datasets across various perturbation levels. Compared to baseline methods, DPPN reduced relative privacy leakage by 5% to 78% while improving downstream utility by 14% to 40%.
- **DPPN achieves comparable performance to white-box defense.** Our black-box neuron detection method performs comparably to a white-box method. On the STS12 dataset, DPPN shows only a 3–6% absolute difference in privacy leakage metrics and less than a 5% relative difference in downstream task performance.
- **Effectiveness against real-world privacy threats.** We test DPPN on two real-world privacy-sensitive datasets: PII-masking 300K and MIMIC-III clinical notes. The results show that DPPN can significantly mitigate the leakage of sensitive information (e.g., sex, disease name) to 17% while baseline methods reduce it only to 43%.

## 2 BACKGROUND

### 2.1 ATTACK SCENARIO

Text embeddings, which are dense vector representations of textual data, pose significant privacy risks due to their ability to inadvertently encode sensitive information Li et al. (2023); Morris et al. (2023). One primary concern is that these embeddings can reveal personal or confidential details present in the input text. In this work, we focus on a specific embedding inversion attack where the adversary aims to reconstruct the input text from the corresponding text embedding. Formally,

given a sequence of text tokens $x$ and the text embedding model $\Phi : x \to \mathbb{R}^d$, where $d$ denotes the embedding dimension, the attacker seeks to find a function $f$ to approximate the inversion function of $\Phi$ as: $f(\Phi(x)) \approx \Phi^{-1}(\Phi(x)) = x$. These inversion attacks can be classified into two categories based on their target: (i) token-level inversion Pan et al. (2020); Song & Raghunathan (2020), which focuses on retrieving individual tokens from the original text, and (ii) sentence-level inversion Li et al. (2023); Morris et al. (2023), which attempts to reconstruct the entire ordered sequence of text. Regardless of the attack model employed, our study prioritizes understanding whether private information (e.g., names, diseases) within the original text is revealed.

## 2.2 PRIVACY-PRESERVING TEXT EMBEDDING

**Privacy Definition.** Preserving privacy is crucial with the rise of powerful pretrained language models. The first step is defining the scope of what constitutes private information. While the concept of privacy can be broad and context-dependent Brown et al. (2022), for practical purposes, a narrower definition is often adopted Sousa & Kern (2023). This definition focuses on personal identifiable information (PII) as privacy concerns, including names, ID numbers, phone numbers, and other similar entities. This definition can extend to named entities in text, such as locations or organizations, depending on the specific privacy requirements of the task.

**Goals.** To clarify the scope of this work, our privacy-preserving text embedding aims to achieve the following two goals:

- *Goal 1 (Defending against sensitive token inference attack)*: For the threat model $\mathcal{A}$ and text embedding $\Phi(x)$, where $x$ is a sentence that contains sensitive information and $\Phi$ is the embedding model. The data owner defines a set of sensitive tokens $T = \{t_1, t_2, \ldots, t_{|T|}\}$ that require to be protected. The objective is to generate an obfuscated embedding $\Phi'(x)$ that prevents the threat model $\mathcal{A}$ from accurately reconstructing or identifying the tokens in $T$.
- *Goal 2 (Maintaining downstream utility)*: The secondary objective is to ensure that the protective measures, while securing the embeddings from inversion attacks, do not compromise the utility of the embeddings in downstream tasks.

**Defender's Knowledge.** Our work primarily addresses a black-box setting, where the defender lacks prior knowledge of the specific attack model employed by adversaries. We focus on developing a robust noise injection mechanism capable of defending against a broad spectrum of inversion attacks without requiring insight into adversarial strategies. However, to comprehensively evaluate our approach, we also explore a white-box scenario in Section 5.1.

## 3 METHODOLOGY

### 3.1 OVERVIEW

We present DPPN, a novel defense framework against embedding inversion attacks. The core concept of our approach is twofold. **Identify privacy neurons**: We employ a differentiable neuron masking learning method to assess the importance of each embedding dimension in carrying token-specific information. The top-$k$ dimensions with the highest importance scores are selected as privacy neurons. **Obfuscate privacy-sensitive information**: We introduce a neuron suppressing perturbation function that adds directional noise to the identified privacy neurons. In constrat to conventional isotropic noise, we show that this perturbation enhances the indistinguishability of embeddings and thus leads to better defense performance. Next, we define the concept of privacy neurons and the associated perturbation framework.

**Definition 1 (Privacy Neurons).** *Consider an input text $x$ and an embedding model $\Phi : x \to \mathbb{R}^d$. We assume there is a subset of dimensions $\mathcal{N}_t \subseteq \mathcal{V} = \{1, \ldots, d\}$ that encapsulates the sensitive information associated with a token $t$. Consequently, the embedding $\Phi(x)$ can be expressed as:*

$$\Phi(x) = (\Phi_{\mathcal{N}_t}(x), \Phi_{\mathcal{V} \setminus \mathcal{N}_t}(x)),$$

*where $\Phi_{\mathcal{N}_t}(x)$ represents the privacy-sensitive neuron activations and $\Phi_{\mathcal{V} \setminus \mathcal{N}_t}(x)$ the privacy-invariant neuron activations.*

For simplicity, we assume the number of privacy neurons (i.e., $|\mathcal{N}_t|$) is a constant $k$ across all tokens. Given the privacy neurons $\mathcal{N}_t$, the data owner shares the perturbed text embedding with:

$$\mathcal{M}(x; \mathcal{N}_t) = \mathcal{F}(\Phi_{\mathcal{N}_t}(x)) \| \Phi_{\mathcal{V} \setminus \mathcal{N}_t}(x), \tag{1}$$

where $\mathcal{F}$ is a randomized perturbation function on selected dimensions $\mathcal{N}_t$, and $\|$ is the concatenation operation.

**Preliminary analysis on privacy-sensitive dimensions.** As described in Eq. 1, embeddings can be decomposed into privacy-sensitive and privacy-invariant activations. To verify the hypothesis, we calculate the dimension-wise sensitivity as: $\Delta_i = \max\left(\{|\Phi(x^+)_i - \Phi(x^-)_i| ; x^+ \in D^+, x^- \in D^-\}\right)$. $\Phi(\cdot)_i$ represents the activation of the $i$-th dimension of the embedding. The sensitivity captures the largest change observed in dimension $i$, with a higher value indicating greater responsiveness to the presence of token $t$. We present a pilot study of the sensitivity distribution of the top and bottom 10% of privacy neurons detected by our approach in Figure 2. Empirically, we found that top privacy neurons exhibit significantly higher sensitivity (average 0.04) compared to tail neurons, whose sensitivity is close to zero. Given this observation, we believe it possible to manipulate only a small portion of dimensions to defend against inversion attacks.

## 3.2 Privacy Neuron Detection through Neuron Masking Leaning

To detect the privacy neurons associated with a sensitive token $t$, we propose a neuron mask learning framework to assess the importance of each neuron. Our objective aims to determine a binary neuron mask, $\mathbf{m} \in \{0, 1\}^d$, which filters irrelevant dimensions and retains the most informative neurons responsible for token $t$ when applied to an embedding $e$. The masked embedding is represented as $e \odot \mathbf{m}$, where $\odot$ denotes the Hadamard product operator. Ideally, a perfect mask would have values of 0 for privacy-irrelevant dimensions and 1 for privacy-related dimensions. However, since the training loss is not differentiable for binary masks, we first introduce a differentiable neuron mask learning framework, followed by a description of the optimization process.

**Differentiable neuron mask learning.** Our goal is to learn a binary mask $\mathbf{m}$ associated with a token $t$, however, the training loss is not differentiable for binary masks. Therefore, we resort to a practical method that employs a smoothing approximation of the discrete Bernoulli distribution Maddison et al. (2017). In our method, we assume each mask $m_i$ follows a hard concrete distribution HardConcrete($\log \alpha_i, \beta_i$) with location $\alpha_i$ and temperature $\beta_i$ Louizos et al. (2018) as:

$$s_i = \sigma\left(\frac{1}{\beta_i}\left(\log\frac{\mu_i}{1 - \mu_i} + \log \alpha_i\right)\right), m_i = \min\left(1, \max\left(0, s_i\left(\xi - \gamma\right) + \gamma\right)\right), \tag{2}$$

where $\sigma$ denotes the sigmoid function. $\xi$ and $\gamma$ are constants, and $\mu_i \sim \mathcal{U}(0, 1)$ is the random sample drawn from the uniform distribution. $\alpha_i$ and $\beta_i$ are learnable parameters. The random variable $s_i$ follows a binary concrete (or Gumbel Softmax) distribution, which is an approximation of the discrete Bernoulli distribution. Samples from the binary concrete distribution are identical to samples from a Bernoulli distribution with probability $\alpha_i$ as $\beta_i \to 0$, and the location $\alpha_i$ allows for gradient-based optimization through reparametrization tricks Jang et al. (2022). During the inference stage, the mask $m_i$ could be derived from a hard concrete gate:

$$m_i = \min\left(1, \max\left(0, \sigma\left(\log \alpha_i\right)\left(\xi - \gamma\right) + \gamma\right)\right). \tag{3}$$

**Learning Objective.** Given a target token $t$ to be protected, we construct a sub-dataset $D^+ = \{x_1, \ldots, x_{|D^+|}\} \subseteq D$ containing sentences with $t$. To measure the embedding change associated with the removal of the token $t$, we create a negative set $D^- = \{\mathcal{R}(x_i, t) \mid x_i \in D^+\}$, where $\mathcal{R}(x_i, t)$ denotes the removal of $t$ from sentence $x_i$. Formally, the objective function could be expressed as:

$$\mathcal{L}(\mathbf{m}, \theta) = -\Sigma_{x^+ \in D^+} \log P_\theta\left(\Phi(x^+) \odot \mathbf{m}\right) - \Sigma_{x^- \in D^-}\left(1 - \log P_\theta\left(\Phi(x^-) \odot \mathbf{m}\right)\right), \tag{4}$$

where $P_\theta(\cdot)$ represents the predicted probability generated by a multi-layer neural network parameterized by $\theta$, and $\Phi(x)$ denotes the embedding of a sentence $x$. To encourage the sparsity, we penalize the $L_0$ complexity of the mask scores by introducing the following regularization term:

$$\mathcal{L}_{reg}(\mathbf{m}) = -\frac{1}{|\mathbf{m}|} \sum_{i=1}^{|\mathbf{m}|} \sigma(\log \alpha_i - \beta_i \log \frac{-\gamma}{\xi}). \tag{5}$$

Finally, we jointly optimize Eq. 4 and Eq. 5. The top-$k$ privacy neurons $\mathcal{N}_t = \text{Top}_k(\mathbf{m})$ are identified by selecting the dimensions with the largest values in $\mathbf{m}$.

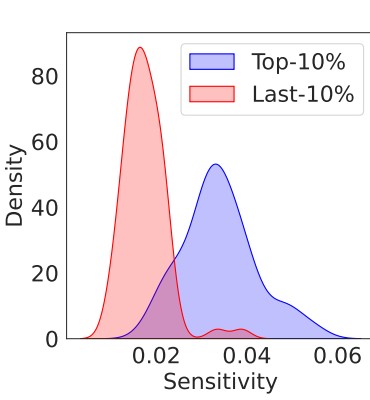

Figure 2: Sensitivity distribution of the top and tail privacy neurons. The Wilcoxon Signed Rank Test indicates a significant difference between the two distributions with a p-value of $1.30e^{-21}$.

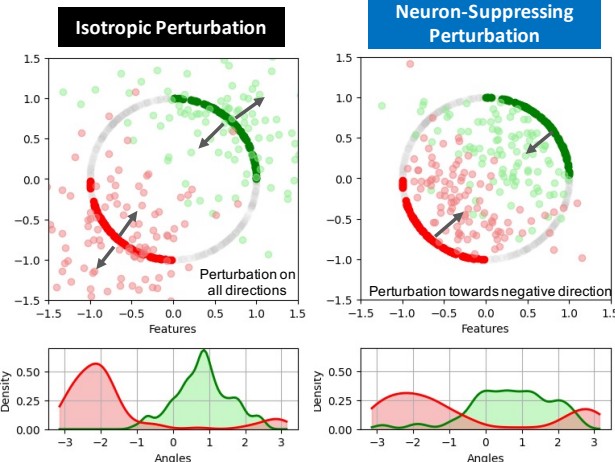

Figure 3: Comparison of isotropic perturbation (left) and neuron-suppressing perturbation (right) in embedding space. **Top:** Scatter plots showing perturbation results, with solid points representing original data and lighter points showing perturbed data. **Bottom:** Corresponding angle distributions.

### 3.3 EMBEDDING PERTURBATION FOR SUPPRESSING PRIVACY INFORMATION

After the privacy neurons are identified, the next step is to perturb these neurons for obfuscating privacy information against the adversary. A common approach is to inject isotropic noise from the Laplace distribution as follows: $\mathbf{e}' = \mathbf{e} + \nu$. Here, $\nu \in \mathbb{R}^d$ is a noise vector with elements $\nu \sim \mathsf{Lap}(0, 1/\epsilon)$ sampled from the Laplace distribution. However, we found that adding isotropic noise fails to effectively obfuscate private information as it perturbs data points towards all directions. Instead of employing the typical DP approach, we propose a novel neuron-suppressing perturbation function that adds random noise that pushes each embedding dimension toward its negative direction. Formally, this can be expressed as:

$$\mathcal{F}(\mathbf{e}) = \mathbf{e} - \mathbf{sign}(\mathbf{e}) \odot \nu'. \tag{6}$$

The perturbation function in Eq. 6 samples one-sided Laplace noise where $\nu'_i = |\nu_i|$ and multiplied by the negative sign of the embedding $\mathbf{e}$. To elucidate the distinction between isotropic noise and neuron-suppressing perturbation, Figure 3 illustrates two distributions represented by red and green data points, along with their perturbed counterparts (in lighter shades) under different perturbation functions. The red and green dots can be conceptualized as the text embeddings sampled from the $D^+$ and $D^-$ datasets in $\mathbb{R}^2$. We also include kernel density estimation (KDE) plots of the angles (i.e., $\arctan2(\mathbf{y}, \mathbf{x})$) below each scatter plot for visualization. As depicted in Figure 3, the isotropic noise applies perturbations in all directions, which is ineffective in obfuscating the data points. In contrast, our proposed neuron-suppressing perturbation introduces noise predominantly in the negative direction that makes the data more indistinguishable. Finally, we apply the perturbation function $\mathcal{F}$ in Eq. 6 and the detected neuron $\mathcal{N}_t$ with Eq. 1 to release the perturbed text embedding.

## 4 EXPERIMENT

### 4.1 EXPERIMENT SETUP

**Datasets.** Our dataset selection aims to address two key objectives: assessing real-world privacy threats and evaluating the privacy-utility tradeoff of defense methods. We utilize PII-Masking-300K Team (2023) and MIMIC-III clinical notes Johnson et al. (2018) to represent distinct real-world threat domains, encompassing 27 Personally identifiable information (PII) classes and medical information, respectively. These datasets, however, do not include specific downstream tasks or labels. To meet the second objective, we select six widely used datasets with downstream labels,

extracting named entities as sensitive information using named entity recognition models. Due to space constraints, we only present results from the STS12 Agirre et al. (2012) and FIQA Maia et al. (2018) datasets in the main experiment, with additional results in Appendix B.

**Attack models.** Three attack models are employed to access the privacy risks of text embedding, including Vec2text Morris et al. (2023), GEIA Li et al. (2023), and MLC Song & Raghunathan (2020). Vec2text and GEIA are sentence-level attack methods that leverage pre-trained GPT models to reconstruct the input sentence. MLC utilizes a three-layer MLP to predict the existence of individual words. Due to its superior performance, Vec2text serves as our primary attack model in subsequent experiments.

**Defense methods.** We compare our DPPN with two noise injection approaches: Laplace mechanism Feyisetan et al. (2020) (LapMech) and Purkayastha mechanism Du et al. (2023) (PurMech). LapMech samples noise from the Laplace distribution, while PurMech utilizes Purkayastha directional noise to perturb embeddings. While these baselines perturb all embedding dimensions and DPPN targets specific dimensions, the perturbation level $\epsilon$ for DPPN is scaled by $\sqrt{k/d}$. This adjustment ensures consistent noise variance with full-dimension methods. In the following experiment, we set $k$ to $d \times 0.2$, selecting the top 20% of privacy neurons as the default configuration.

**Evaluation metrics.** We evaluate privacy leaks in text embeddings using two metrics: Leakage and Confidence. Leakage measures the attack model's accuracy in predicting sensitive tokens, with lower values indicating better defense. Confidence represents the attack model's maximum probability of predicting sensitive tokens, where lower values suggest reduced likelihood of generating target sensitive information. As an indicator for downstream utility, we report dataset-specific downstream performance as our utility metric.

**Embedding models.** Following the research by previous works Morris et al. (2023); Huang et al. (2024), we include three widely used embedding models: GTR-base Ni et al. (2022b), Sentence-T5 Ni et al. (2022a), and SBERT Reimers & Gurevych (2019) to validate the robustness of DPPN. GTR-base is used by default due to its higher vulnerability to the Vec2text attack.

Table 1: Privacy-utility tradeoff across various defense methods. Privacy leakage is evaluated using the Leakage and Confidence metrics, with lower values indicating stronger privacy protection. Utility is measured by the downstream performance on specific data tasks. The mean and standard deviation of 5 runs are reported in percentages(%).

| Dataset | $\epsilon$ | Privacy Metrics | | | | | | Utility Metric | | |
| --- | --- | --- | --- | --- | --- | --- | --- | --- | --- | --- |
| | | Leakage ↓ | | | Confidence ↓ | | | Downstream ↑ | | |
| | | LapMech | PurMech | DPPN | LapMech | PurMech | DPPN | LapMech | PurMech | DPPN |
| STS12 | 1 | 7.36 ±0.61 | 7.42 ±0.49 | **1.61** ±0.16 | 6.70 ±0.32 | 6.80 ±0.29 | **6.05** ±0.31 | 29.28 ±0.00 | 29.31 ±0.00 | **40.78** ±0.00 |
| | 2 | 22.34 ±1.38 | 22.66 ±1.15 | **13.44** ±0.60 | 9.39 ±0.17 | 9.42 ±0.17 | **8.25** ±0.34 | 60.72 ±0.00 | 60.72 ±0.00 | **67.05** ±0.00 |
| | 4 | 38.17 ±0.86 | 38.04 ±0.71 | **33.49** ±0.67 | 24.70 ±0.75 | 24.74 ±0.71 | **23.80** ±0.55 | 72.47 ±0.00 | 72.47 ±0.00 | **73.40** ±0.00 |
| | 6 | 44.74 ±0.43 | 44.76 ±0.49 | **42.59** ±0.82 | 34.59 ±0.32 | 34.59 ±0.24 | **34.14** ±0.67 | 73.68 ±0.00 | 73.68 ±0.00 | **73.95** ±0.00 |
| | 8 | 48.48 ±0.60 | 48.34 ±0.57 | **47.11** ±0.66 | 38.75 ±0.80 | 38.82 ±0.79 | **38.49** ±0.76 | 73.98 ±0.00 | 73.98 ±0.00 | **74.09** ±0.00 |
| | $\infty$ | 60.09 | | | 47.81 | | | 74.25 | | |
| FIQA | 1 | 12.56 ±0.98 | 13.01 ±1.40 | **2.01** ±0.22 | 6.67 ±0.51 | 6.70 ±0.49 | **5.84** ±0.33 | 10.64 ±0.24 | 10.63 ±0.25 | **15.05** ±0.31 |
| | 2 | 35.17 ±1.46 | 35.31 ±0.86 | **20.15** ±1.34 | 16.70 ±0.74 | 16.55 ±0.66 | **11.92** ±0.62 | 21.74 ±0.36 | 21.76 ±0.29 | **25.96** ±0.33 |
| | 4 | 55.69 ±1.05 | 55.38 ±1.26 | **51.26** ±1.18 | 35.32 ±0.74 | 35.25 ±0.78 | **31.36** ±0.63 | 32.22 ±0.14 | 32.23 ±0.13 | **32.84** ±0.23 |
| | 6 | 64.12 ±0.82 | 64.13 ±0.85 | **62.79** ±1.71 | 43.35 ±1.50 | 43.56 ±1.53 | **41.57** ±1.41 | 33.24 ±0.03 | 33.26 ±0.04 | **33.58** ±0.13 |
| | 8 | 68.85 ±1.26 | 68.63 ±1.36 | **67.99** ±0.50 | 48.07 ±1.08 | 47.77 ±0.78 | **46.25** ±0.86 | 33.50 ±0.14 | 33.52 ±0.15 | **33.73** ±0.10 |
| | $\infty$ | 77.35 | | | 54.48 | | | 33.56 | | |

## 4.2 Privacy-Utility Trade-off Analysis

To evaluate the privacy-utility tradeoff among different defense methods and privacy levels, we present experimental results for the STS12 and FIQA datasets in Table 1. Note that $\epsilon = \infty$ represents the non-protected embedding. Our findings demonstrate that DPPN exhibits a superior privacy-utility tradeoff compared to baseline methods such as LapMech and PurMech. For instance, with the STS12 dataset at $\epsilon = 2$, DPPN reduces Leakage from 60% (unprotected) to 13%, while baseline methods only achieve a reduction to 22%. Importantly, while DPPN effectively mitigates privacy leakage, it also maintains or enhances the downstream performance relative to baseline methods. It is worth noting that the baseline methods perturb all embedding dimensions as a general defense

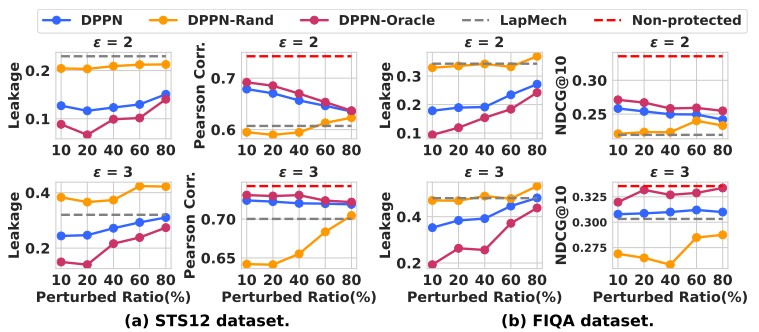

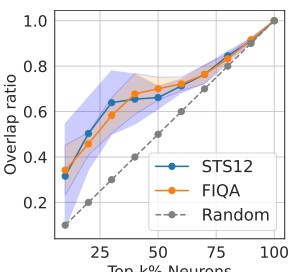

Figure 5: Top-$r\%$ neuron overlap ratio calculated between black-box and white-box detected neurons.

Figure 4: Comparison of different privacy neuron detection methods under various perturbation neuron ratios and perturbation levels of $\epsilon$.

mechanism. However, the results for DPPN suggest that selectively perturbing specific privacy neurons could be a more effective approach when the goal is to protect a particular privacy concept without compromising downstream utility.

## 5 FURTHER DISCUSSION ON PRIVACY NEURONS

### 5.1 EVALUATION ON NEURON DETECTION METHODS

Our work rests on the fundamental assumption that privacy neurons can be detected and perturbed to defend against inversion attacks. This premise raises two critical questions: (i) How effectively can we defend against inversion attacks using ground truth privacy neurons? and (ii) To what extent can our black-box detection model approach this ideal defense? To address this, we examine a white-box defense scenario where the defender possesses complete knowledge of the attack model's parameters.

**White-box privacy neuron detection.** Under white-box access to the attack model, we employ the Fast Gradient Sign Method (FGSM) Goodfellow et al. (2014) to identify the most influential neurons for privacy protection. Our approach involves computing the gradient of the attack model's loss with respect to the input text embedding for a specific sensitive token. Neurons with the highest average gradient magnitudes are identified as privacy neurons. This white-box defense method is referred to as DPPN-Oracle in subsequent experiments.

**Comparison of neuron detection methods.** Figure 4 presents experimental results evaluating various privacy neuron detection methods, including our black-box method (DPPN), the white-box approach (DPPN-Oracle), and a random selection method (DPPN-Rand), alongside LapMech and non-protected baselines for reference. The white-box method consistently achieves the best privacy-utility tradeoff, confirming that perturbing the most informative neurons significantly reduces privacy leakage. Notably, our black-box method performs comparably to the white-box approach; at $\epsilon = 2$, it exhibits an absolute Leakage difference of only 3% to 6%, with less than a 5% relative difference in downstream metrics. In contrast, the random selection method is significantly less effective. Furthermore, Figure 5 depicts the top-$r\%$ overlap ratio between the black-box and white-box neurons. The results indicate that our black-box detection methods successfully identify neurons with 32% and 51% accuracy for the top 10% and 20% of neurons, respectively. To conclude, these results demonstrate the effectiveness of DPPN in approximating the ideal white-box scenario.

### 5.2 PERTURBATION FUNCTIONS ON PRIVACY NEURONS

A key component of DPPN is the use of the neuron-suppressing perturbation function. To evaluate the impact of various perturbation functions applied to privacy neurons, Table 2 presents results for applying these functions across all dimensions ($r = 100\%$) and to the top-$r\%$ privacy neurons. We found that isotropic perturbation functions like LapMech and PurMech have minimal impact on performance when perturbing privacy neurons. For example, applying LapMech to perturb 10% and 20% of privacy neurons results in slight increases in leakage by 0.48% and 1.16%, respectively. A similar trend is observed with PurMech. This limited impact could be attributed to the weak

Table 2: Defense and downstream performance using different perturbation functions with $\epsilon = 2$. We vary the ratio $r$ to select the top-$r\%$ sensitive neurons detected by DPPN. We report all evaluation metrics in percentage (%). The relative improvement compared to the full perturbation is reported within the parentheses.

| Perturb. Ratio | Full ($r = 100\%$) | | $r = 10\%$ | | $r = 20\%$ | |
|---|---|---|---|---|---|---|
| Perturb. Function | Leakage ↓ | Downstream ↑ | Leakage ↓ | Downstream ↑ | Leakage ↓ | Downstream ↑ |
| LapMech | 14.53 | 45.95 | 14.60 (+0.48%) | 45.93 (-0.04%) | 14.70 (+1.16%) | 45.85 (-0.22%) |
| PurMech | 14.33 | **45.97** | 14.57 (+1.67%) | 45.94 (-0.07%) | 14.60 (+1.88%) | 45.87 (-0.22%) |
| Suppress (Ours) | **8.29** | 40.69 | **7.01 (-15.44%)** | 59.05 (+45.12%) | **5.45 (-34.26%)** | 56.97 (+40.01%) |

perturbation as illustrated in Figure 3. In contrast, our suppress method yields significant reductions in leakage and notable improvements in downstream performance. Specifically, at $r = 10\%$, leakage decreases by 15.44%, and downstream performance enhances by 45.12%.

### 5.3 QUALITATIVE ANALYSIS ON DETECTED NEURONS

We present a qualitative analysis to examine the quality of privacy neurons identified by DPPN for individual words, as visualized in Figure 6. We selected six groups of semantically similar words: weekdays, countries, months, USA-related terms, gender-related terms, and numbers. The x-axis displays the union of the top-5 neuron indices associated with each word. We have the following two findings: **1) Semantic similar words share similar privacy neurons.** As depicted in Figure 6, we found that words with similar semantics, such as weekdays or countries, tend to cluster around the same neuron dimensions. This indicates that the privacy neurons identified by DPPN effectively capture contextually relevant and meaningful information. **2) DPPN provides implicit protection on semantically similar words.** Given the pre-

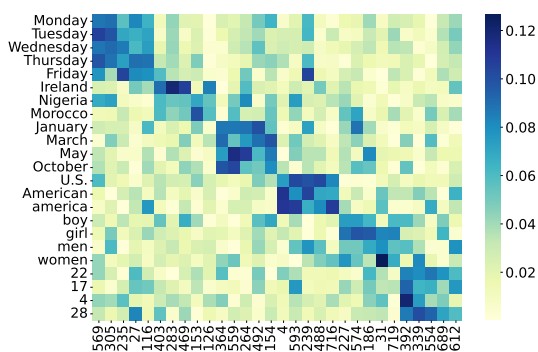

Figure 6: Visualization of the neuron mask for individual tokens, where larger weights represent higher neuron importance.

vious finding, when DPPN suppresses privacy neurons for a specific word, it implicitly extends protection to semantically related words. As shown in Table 9 in the Appendix, we calculate the indirect leakage performance to assess the level of implicit protection. For semantically similar tokens, the leakage mitigation rate reaches up to 36% to 46%, while for other unrelated tokens only reduces by 11% to 29%.

Table 3: Defense performance $w.r.t.$ different attack models. We report the leakage metric in percentage (%) on the STS12 dataset. In addition, we highlight the relative performance compared to non-protected in red.

| Attack Models | $\epsilon = \infty$ | $\epsilon = 1$ | | | $\epsilon = 2$ | | |
|---|---|---|---|---|---|---|---|
| | | LapMech | PurMech | DPPN | LapMech | PurMech | DPPN |
| Vec2text Morris et al. (2023) | 60.09 | 6.94 (-88.45%) | 7.05 (-88.27%) | **1.29 (-97.85%)** | 22.97 (-61.77%) | 22.39 (-62.74%) | **11.65 (-80.61%)** |
| GEIA Li et al. (2023) | 25.34 | 12.30 (-51.46%) | 12.36 (-51.22%) | **7.08 (-72.06%)** | 20.60 (-18.71%) | 21.21 (-16.30%) | **15.82 (-37.57%)** |
| MLC Song & Raghunathan (2020) | 53.20 | 49.39 (-7.16%) | 49.80 (-6.39%) | **47.63 (-10.47%)** | 52.74 (-0.86%) | 52.68 (-0.97%) | **49.59 (-6.79%)** |

## 6 ROBUSTNESS ANALYSIS OF DPPN

### 6.1 DEFENDING AGAINST DIFFERENT ATTACK MODELS

Given that our privacy neuron detection process is attack-model agnostic, it is crucial to evaluate the robustness of DPPN across various adversarial scenarios. We evaluated the defense capabilities of DPPN against three distinct attack methods: MLC Song & Raghunathan (2020), GEIA Li et al. (2023), and Vec2text Morris et al. (2023). As shown in Table 3, DPPN consistently outperforms

LapMech and PurMech across all attack models by a significant margin. Our findings reveal that complex attack models, such as Vec2text and GEIA, are more susceptible to embedding perturbation, exhibiting substantial leakage reductions of 88% and 51% respectively at $\epsilon = 1$. In contrast, the shallow MLC model demonstrates less vulnerability to our defense method. These experimental results validate the efficacy of DPPN in mitigating information leakage across diverse adversarial settings.

Table 4: Defense performance on different categories of sensitive information. We report the leakage metric in percentage (%) with $\epsilon = 2$.

| Dataset | PII-300K | | | | MIMIC-III | | | | STS12 | | |
|---|---|---|---|---|---|---|---|---|---|---|---|
| Category | Sex | City | State | Country | Age | Sex | Disease | Symptom | Name | Location | Random |
| Non-protected | 86.12 | 68.45 | 75.43 | 84.07 | 58.49 | 88.40 | 70.43 | 82.76 | 62.20 | 49.69 | 78.24 |
| LapMech | 42.35 | 33.39 | 36.63 | 40.37 | 31.88 | 43.38 | 23.32 | 38.17 | 21.60 | 16.15 | 49.01 |
| PurMech | 43.53 | 34.10 | 38.45 | 41.45 | 31.89 | 43.38 | 22.86 | 31.30 | 20.73 | 15.53 | 49.24 |
| DPPN | **28.24** | **15.13** | **21.47** | **24.21** | **25.91** | **17.43** | **15.57** | **26.83** | **4.18** | **8.70** | **35.29** |

## 6.2 EFFECTIVENESS AGAINST REAL-WORLD PRIVACY THREATS

We evaluated DPPN's resilience to inversion attacks across various data domains and privacy categories. This evaluation used the PII-Masking 300K dataset Team (2023), MIMIC-III clinical notes Johnson et al. (2018), and the STS12 datasets. For STS12, we selected 100 random words to represent a broad spectrum of privacy scenarios. Table 4 presents our experimental results. The results show the significant vulnerability of unprotected embeddings to inversion attacks. For example, in the MIMIC-III dataset, the attack model inferred sensitive information with high accuracy. It achieved 88% accuracy for sex, 70% for diseases, and 82% for symptoms. In contrast, DPPN shows a notable improvement in reducing privacy leaks compared to existing methods. With the same level of perturbation, DPPN lowers sex information leakage from 88% to 17%. Meanwhile, LapMech and PurMech remain much higher at 43%. This trend is consistent across other privacy categories.

Table 5: Defense and downstream performance $w.r.t.$ different embedding models under $\epsilon = 2$. We use STS12 dataset and report the mean and standard deviation of 5 runs for all evaluation metrics.

| Embedding Models | GTR-base | | Sentence-T5 | | SBERT | |
|---|---|---|---|---|---|---|
| Metrics | Leakage ↓ | Downstream ↑ | Leakage ↓ | Downstream ↑ | Leakage ↓ | Downstream ↑ |
| Non-protected | 60.09 | 74.25 | 43.83 | 86.79 | 42.11 | 81.36 |
| LapMech | 22.66 ±0.62 | 60.72 ±0.00 | 31.71 ±0.62 | 63.16 ±0.00 | 23.82 ±0.89 | 77.89 ±0.00 |
| PurMech | 22.88 ±0.67 | 60.72 ±0.00 | 32.11 ±0.47 | 63.15 ±0.00 | 23.59 ±0.78 | 77.89 ±0.00 |
| DPPN | **13.11** ±0.81 | **67.05** ±0.00 | **22.38** ±0.44 | **74.45** ±0.00 | **17.15** ±0.74 | **79.42** ±0.00 |

## 6.3 DEFENSE PERFORMANCE ON VARIOUS EMBEDDING MODELS

While previous experiments utilized GTR-base as the default embedding model, Table 5 extends the evaluation to two additional embedding models to validate the robustness of DPPN. When using LapMech and PurMech perturbations, leakage is reduced to approximately 20% to 30%, with downstream performance dropping to 60% to 70%. In contrast, DPPN reduces leakage to 13% with GTR-base and 17% with SBERT while preserving higher downstream performance compared to other defense methods. The results verify that the effectiveness of DPPN is consistent regardless of the embedding models.

## 7 CASE STUDY ON MIMIC-III DATASET

To demonstrate the privacy risks in a specific threat domain, we conducted a case study using MIMIC-III clinical notes Johnson et al. (2018). Table 6 presents the results of embedding inversion attack on two types of sensitive tokens ("age" and "disease name") with different noise levels. We assessed the semantic fidelity of the reconstructed sentences by comparing their similarity to the original text using cosine similarity from an external embedding model.

Table 6: Case study on the MIMIC-III dataset with two sensitive words and perturbation level $\epsilon$. We highlight the leakage of sensitive words and demonstrate the semantic similarity of the reconstructed sentence to the ground truth.

| **Example 1: Protect age with strong noise $\epsilon = 1$** | | | |
|---|---|---|---|
| Method | Defense | Semantic | Reconstructed Sentence |
| Ground truth | - | - | this **68-year-old** white male has a history of diabetes, hyperlipidemia and hypertension |
| Non-private | **Failed** | 0.98 | this **68-year-old** white male has a history of hypertension, hyperlipidemia, and diabetes. |
| LapMech | **Success** | 0.11 | age (e.g., blood edemas in males of African PH whose history has been hyperesoteric |
| PurMech | **Success** | 0.11 | age (e.g., blood edemas in males of African PH whose history has been hyperesoteric |
| DPPN | **Success** | 0.62 | a white male with diabetes has existing Hyperlipidemia history |
| **Example 2: Protect disease name with weak noise $\epsilon = 2$** | | | |
| Ground truth | - | - | this male has had known **coronary** disease and prior silent myocardial infarction. |
| Non-private | **Failed** | 0.95 | this male has known silent **coronary** disease and has had prior myocardial infarction. |
| LapMech | **Failed** | 0.23 | male has known **coronary** myopathy. Silent rib syndrome, white-fiddled gyne, and ca |
| PurMech | **Failed** | 0.18 | male has known **coronary** myopathy. Silent-fidged heart attacks. White-fidged-fid |
| DPPN | **Success** | 0.54 | an active male with myocardial infarction, congestive heart disease. |

In Example 1, we applied a strong perturbation level of $\epsilon = 1$ to perturb the text embeddings. Under this condition, all three defense methods (LapMech, PurMech, and DPPN) effectively prevented the leakage of sensitive age information. However, LapMech and PurMech significantly degraded the semantic quality of the embeddings with only 11% of the original semantic similarity. In contrast, DPPN maintained 62% semantic similarity. In Example 2, we used a lower perturbation level of $\epsilon = 2$. Here, both LapMech and PurMech failed to protect against privacy leakage and further compromised the semantic integrity of the embeddings. Conversely, DPPN successfully safeguarded the sensitive information while preserving semantic quality of the embeddings.

## 8 RELATED WORK

**Inversion attacks on embeddings.** Embedding inversion attacks pose significant privacy risks in both computer vision and natural language processing (NLP). These attacks exploit the unintended memorization capabilities of neural models, allowing adversaries to reconstruct original data from embeddings. In computer vision, high-fidelity reconstructions of images from embeddings have been demonstrated Bordes et al. (2022); Dosovitskiy & Brox (2016); Teterwak et al. (2021). Similarly, in NLP, embeddings can reveal sensitive text data and even demographic information about authors Pan et al. (2020); Song & Shmatikov (2019); Lyu et al. (2020); Coavoux et al. (2018). The recent work Morris et al. (2023) shows that embeddings from services like OpenAI's can be accurately inverted to recover the original text.

**Privacy-preserving text embeddings.** Two lines of work were explored for generating privacy-preserving text embeddings: adversarial training and noisy embedding. The noisy embedding approach defends against inversion attacks by adding random noise to the embeddings. For instance, Laplace noise has been widely used to defend against inversion attacks Morris et al. (2023), membership inference Song & Raghunathan (2020), and attribute inference attacks Coavoux et al. (2018). Advanced techniques like the Purkayastha mechanism Du et al. (2023) further enhances the Laplace method for superior defense performance against inference attacks. On the other hand, adversarial training Coavoux et al. (2018); Elazar & Goldberg (2018); Li et al. (2018) involves creating a simulated adversary that tries to infer sensitive information while the main model is optimized to confuse this adversary. However, this approach's success largely depends on the quality of the simulated adversary Zhang et al. (2018).

## 9 CONCLUSION

In this work, we addressed the privacy risks of text embeddings, particularly against embedding inversion attacks. By identifying and suppressing privacy neurons, our method enhances defense with minimal impact on downstream tasks. Extensive experiments validate the effectiveness of DPPN across various attack models, embedding models, and real-world privacy threats. As privacy risks grow and attack models advance, we aim for our work to establish a robust framework for protecting sensitive information in text embeddings.

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

## A  LIMITATIONS

While DPPN demonstrates significant effectiveness in protecting sensitive information within text embeddings, it lacks the theoretical guarantees provided by differential privacy (DP). DP offers a formal framework to quantify privacy, ensuring that the inclusion or exclusion of a single data point does not significantly affect an algorithm's output. This gap between DPPN and DP is particularly relevant for applications involving highly sensitive data, such as medical records or financial information.

The primary challenge in bridging this gap lies in adapting DPPN's targeted perturbation approach to meet DP's rigorous standards. DP requires that all released data be perturbed to provide a strong, provable privacy guarantee. However, DPPN only perturb a subset of dimensions while keeping the remaining dimensions intact. To bridge this gap, future research could explore developing a hybrid approach that applies DP-compliant noise to all dimensions while concentrating higher magnitude perturbations on privacy-sensitive neurons such as Mahalanobis mechanism Xu et al. (2020) or Rényi differential privacy Mironov (2017).

## B  COMPLETE DEFENSE PERFORMANCE ACROSS ALL DATASETS

In addition to the STS12 Agirre et al. (2012) and FIQA Maia et al. (2018) datasets used in the main experiment, Table 8 also presents statistics of other datasets, including STSB Cer et al. (2017), STS14 Agirre et al. (2014), Quora Bondarenko et al. (2020), and NFCorpus Boteva et al. (2016). Figure 7 and Figure 8 show the complete defense performance on all datasets. Besides using Leakage, we also utilize Confidence to assess the defense performance. This metric reflects the certainty of the attack model's predictions. A higher Confidence score indicates that the model is more confident in its prediction of the sensitive token. For the semantic textual similarity (STS) task, downstream performance is measured using the Pearson correlation of Cosine Similarity (Pearson corr.). In the context of information retrieval, we employ the ranking metric NDCG@10. As described in Section 4.2, DPPN consistently demonstrates superior performance over LapMech and PurMech across all levels of perturbation and datasets, both in defense and downstream task metrics.

## C  EXTRACTING SENSITIVE WORDS

MIMIC-III clinical notes Johnson et al. (2018) is an anonymous electronic health record database containing extensive clinical data from intensive care units. We use the biomedical Named Entity Recognition (NER) model Raza et al. (2022) to extract privacy-related medically named entities including age, sex, disease, and symptom. For the STS12 dataset, name and location are extracted by leveraging the named entity recognition tool from the Spacy library[2]. In addition, we select 100 random words to represent broader privacy scenarios.

## D  IMPLEMENTATION DETAILS OF ATTACK MODELS

To better measure defense performance, we load pre-trained vec2text[3] and fine-tuned attack models for 50 epochs individually for different perturbation methods, specifically LapMech, PurMech, and DPPN. This approach simulates the scenario where attackers train their models and allows for a comprehensive assessment of Leakage and Confidence.

---

[2] https://github.com/explosion/spacy-models/releases/tag/en_core_web_sm-3.7.0

[3] https://huggingface.co/ielabgroup/vec2text_gtr-base-st_inversion

Table 7: Privacy-utility tradeoff across different defense Methods. Privacy leakage is assessed using Leakage and Confidence metrics, where lower values indicate stronger privacy protection. Utility is measured by data-specific downstream performance. All metrics are presented as percentages (%).

| Dataset | $\epsilon$ | Privacy Metrics | | | | | | Utility Metric | | |
|---|---|---|---|---|---|---|---|---|---|---|
| | | Leakage ↓ | | | Confidence ↓ | | | Downstream ↑ | | |
| | | LapMech | PurMech | DPPN | LapMech | PurMech | DPPN | LapMech | PurMech | DPPN |
| STSB | 1 | 20.75 | 19.03 | **2.68** | 1.89 | 1.98 | **1.78** | 40.03 | 40.05 | **48.17** |
| | 2 | 53.79 | 49.95 | **32.39** | 15.40 | 13.97 | **8.07** | 71.87 | 71.85 | **76.14** |
| | 4 | 71.82 | 69.15 | **64.79** | 41.76 | 38.44 | **36.33** | 80.95 | 80.95 | **81.00** |
| | 6 | 76.15 | 74.98 | **73.10** | 52.36 | 49.28 | **48.63** | 81.08 | 81.08 | 80.91 |
| | 8 | 78.94 | 77.40 | **76.42** | 56.84 | 54.02 | **53.98** | 80.95 | 80.95 | 80.81 |
| | $\infty$ | 86.75 | | | 66.57 | | | 80.64 | | |
| STS14 | 1 | 1.03 | 1.55 | **0.30** | 20.42 | 20.88 | **18.05** | 39.76 | 39.71 | **48.47** |
| | 2 | 4.04 | 4.13 | **2.41** | 21.86 | 21.84 | **21.10** | 70.28 | 70.25 | **74.44** |
| | 4 | **8.64** | 8.77 | 9.46 | 28.05 | **27.73** | 28.56 | 79.16 | 79.16 | **79.31** |
| | 6 | **11.22** | 11.26 | 14.70 | **30.38** | 30.39 | 32.95 | **79.47** | **79.47** | 79.37 |
| | 8 | 13.67 | **13.50** | 16.12 | 32.09 | **32.05** | 34.81 | **79.43** | **79.43** | 79.32 |
| | $\infty$ | 21.97 | | | 35.99 | | | 79.25 | | |
| Quora | 1 | 25.96 | 25.87 | **2.85** | 2.71 | 2.70 | **1.57** | 11.89 | 11.78 | **72.33** |
| | 2 | 57.44 | 54.78 | **33.67** | 18.62 | 15.92 | **9.94** | 70.04 | 70.19 | **82.19** |
| | 4 | 75.56 | 75.80 | **68.00** | 50.87 | 51.00 | **41.21** | 82.79 | 82.75 | **83.94** |
| | 6 | 81.65 | 81.65 | **76.75** | 58.99 | 59.08 | **53.43** | 83.70 | 83.72 | **84.02** |
| | 8 | 83.69 | 83.64 | **79.79** | 62.28 | 62.06 | **57.32** | 83.90 | 83.91 | **83.97** |
| | $\infty$ | 89.30 | | | 68.30 | | | 84.01 | | |
| NFCorpus | 1 | 7.77 | 8.45 | **0.68** | 1.27 | 1.06 | **0.83** | **23.70** | 23.61 | 19.94 |
| | 2 | 29.73 | 31.42 | **12.16** | 15.92 | 15.51 | **6.73** | 27.31 | 27.38 | **29.61** |
| | 4 | 56.76 | 55.41 | **46.96** | 45.70 | 46.26 | **35.36** | 30.76 | 30.75 | **31.04** |
| | 6 | 69.93 | 69.26 | **57.77** | 58.27 | 58.00 | **48.09** | 31.32 | 31.32 | **31.37** |
| | 8 | 78.72 | 79.05 | **66.55** | 63.89 | 63.83 | **53.85** | **31.56** | **31.56** | 31.52 |
| | $\infty$ | 88.18 | | | 75.54 | | | 31.63 | | |

| Dataset | STS12 | FIQA | STSB | STS14 | Quora | NFCorpus | MIMIC-III | PII-300K |
|---|---|---|---|---|---|---|---|---|
| Downstream task | STS | Retrieval | STS | STS | Retrieval | Retrieval | - | - |
| Domain | SemEval | Financial | SemEval | SemEval | QA | Medical | Medical | PII |
| Sentences | 10684 | 5500 | 17256 | 3000 | 10000 | 2590 | 4244 | 177677 |
| Average sentence length | 14.53 | 10.80 | 10.17 | 9.77 | 9.53 | 3.31 | 15.03 | 47.12 |
| Unique named entities | 123 | 41 | 228 | 41 | 90 | 13 | 290 | 491 |
| Evaluation metric | Pearson Corr. | NDCG@10 | Pearson Corr. | Pearson Corr. | NDCG@10 | NDCG@10 | - | - |

Table 8: Statistics of datasets.

Table 9: Leakage mitigation rate (%) compared to non-protected embeddings. We report the leakge metric using DPPN with $\epsilon = 2$.

| | Weekdays | Country | City |
|---|---|---|---|
| **Target tokens** | -76.2% | -64.3% | -72.5% |
| **Semantic similar tokens** | -46.2% | -36.2% | -42.8% |
| **Other tokens** | -11.7% | -29.1% | -12.6% |

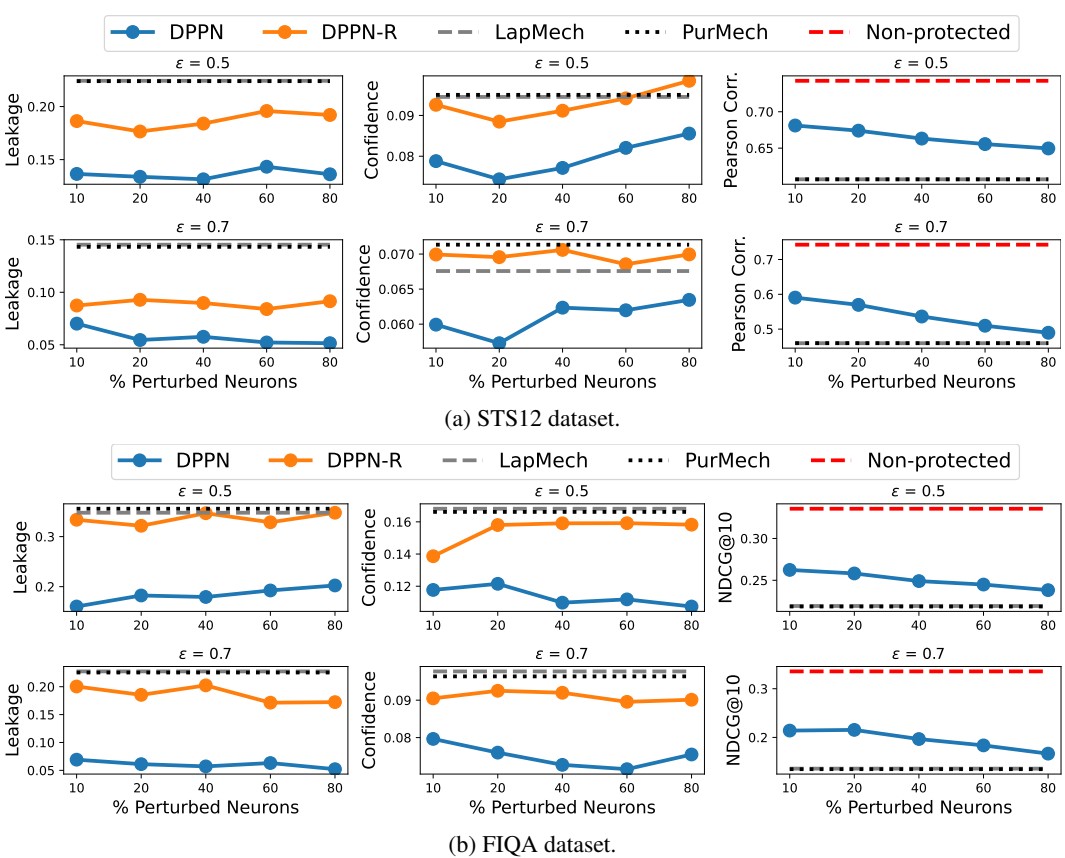

Figure 7: Privacy-utility tradeoff across various datasets with different perturbation levels of $\epsilon$. For each dataset, the left and middle sections evaluate defense effectiveness through Leakage and Confidence metrics, where lower values indicate better defense. The right section illustrates downstream performance, where a higher score is better.

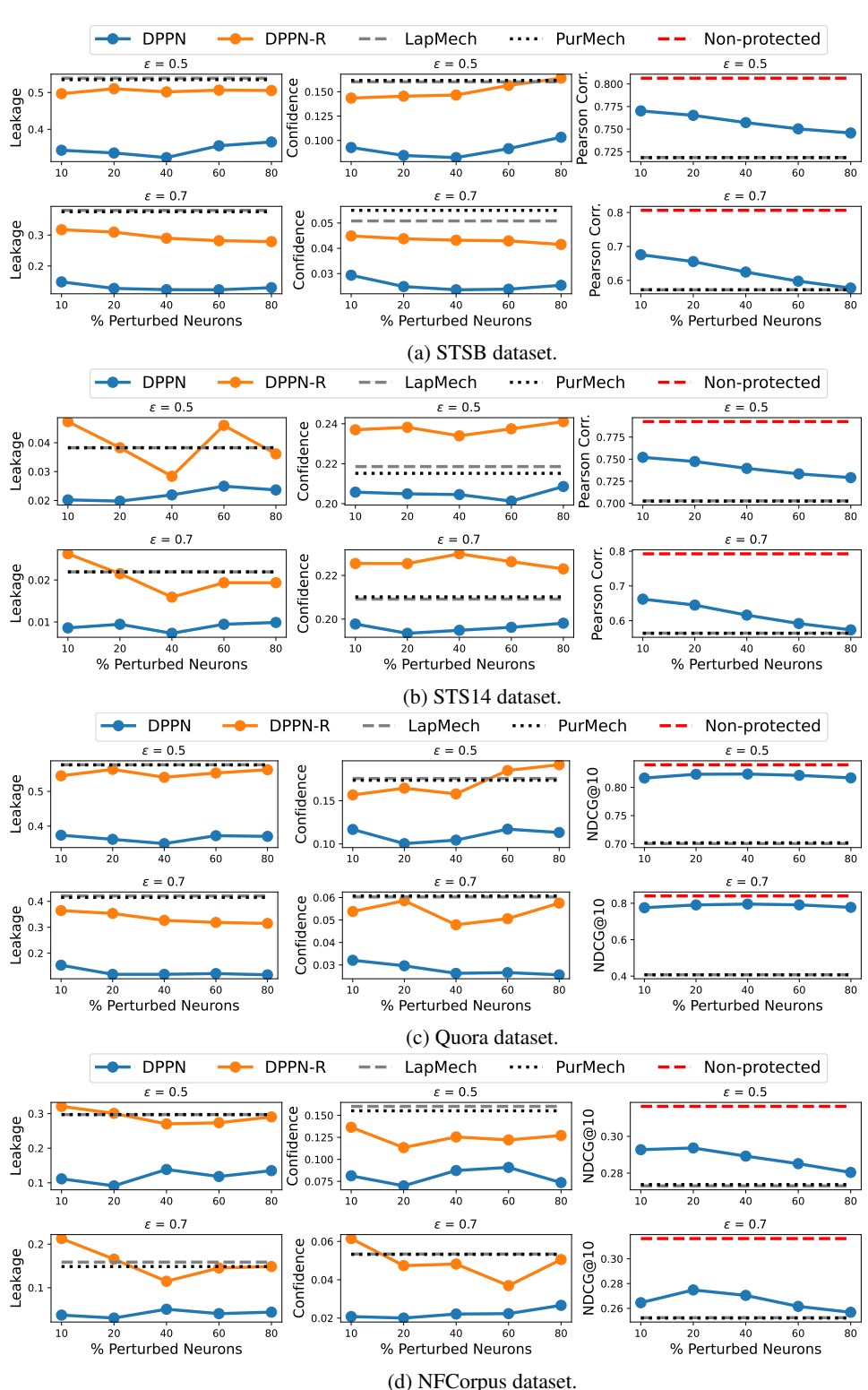

Figure 8: Privacy-utility tradeoff across various datasets with different perturbation levels of $\epsilon$. For each dataset, the left and middle sections evaluate defense effectiveness through Leakage and Confidence metrics, where lower values indicate better defense. The right section illustrates downstream performance, where a higher score is better.

