# OpenReview forum: "Detecting and Perturbing Privacy-Sensitive Neurons to Defend Embedding Inversion Attacks"
_ICLR.cc/2025/Conference — Submitted to ICLR 2025_

### Official Review · Reviewer_FFaT · 2024-10-17

**Soundness:** 3
**Presentation:** 2
**Contribution:** 3
**Rating:** 6
**Confidence:** 3

**Summary:**

The paper proposes a defense method DPPN to resist against embedding inversion attacks. In contrast to previous methods that add noise on all embedding dimensions, it recognize privacy neurons that contains more sensitive information and only perturbs them. Therefore, it achieve an excellent privacy-utility trade-off. Extensive experiments show that the defense method can protect private information while maintaining the origin semantic information.

**Strengths:**

Originality: The introduction of privacy neurons and targeted perturbation is innovative, departing from conventional methods that apply noise to all dimensions.

Quality: The conducted experiments are comprehensive. The study evaluates DPPN across six datasets, multiple embedding models, and various attack models, showcasing robust and thorough experimental design.

Clarity: The paper follows a clear structure, with a logical flow from problem motivation to solution, experiments, and results.

Significance: The ability to reduce privacy leakage without sacrificing utility makes DPPN relevant for real-world applications where maintaining both privacy and accuracy is critical.

**Weaknesses:**

+ The paper does not provide sufficient detail on how parameters $\xi$ and $\eta$ in formulas 3 and 5 are selected during the experiments. Please provide additional information about the selection process. This clarification would help readers understand their impact and ensure reproducibility.
+ The concepts of $D^+$ and $D^-$ are introduced on line 170, but their explanations are deferred until line 204. This gap may confuse readers. It would be clearer if a brief definition were provided when these concepts are first mentioned, or if the detailed explanation were moved closer to the first appearance.
+ In Fig. 2, the paper introduces the concept of sensitivity for privacy neurons. What if privacy neurons were selected based on their top dimension-wise sensitivity? The paper lacks a study on it.

**Questions:**

+ The paper uses a fixed top-k selection method for privacy neurons. What if we choose them based on a threshold of $m$, such as 0.5?

---

> ### Author Response · Authors · 2024-11-21
>
> We thank the reviewer for their feedback and will respond to the raised questions below.
>
> **W1: The paper does not provide sufficient detail on how parameters $\xi$ and $\gamma$ in formulas 3 and 5 are selected during the experiments.**
>
> Thank you for pointing out the missing details regarding the parameter selection of $\xi$ and $\gamma$ in Formulas 3 and 5. We followed the optimal settings established in prior work [1], using $\xi = 1.1$ and $\gamma = -0.1$. We will incorporate this clarification into the revised manuscript to enhance transparency and reproducibility.
>
>
> **W2: What if privacy neurons were selected based on their top dimension-wise sensitivity?**
>
> We appreciate the reviewer’s suggestion and agree that selecting privacy neurons based on their top dimension-wise sensitivity could serve as a meaningful baseline for neuron selection methods. To evaluate this approach, we conducted experiments using sensitivity-based selection and compared the results with DPPN and random selection.
> Here we report the privacy metric (Leakage) and utility metric (Downstream) by varying the top-$r$% privacy neurons. The results are presented in the tables below.
>
> The results indicate that while sensitivity-based selection is effective in mitigating privacy leakage compared to random selection, DPPN demonstrates significantly superior performance. Specifically, the privacy neurons identified by DPPN result in lower leakage and higher downstream utility across different settings. This improvement can be attributed to DPPN's ability to assign higher weights to critical neurons via the neuron mask learning process. We hope the results clarify the reviewer's concern.
>
> **STS12 dataset**
> | **Ratio** | **Leakage $\downarrow$(%)** | || **Downstream $\uparrow$(%)** | ||
> | -------- | -------- | -------- | -------- | -------- |-------- | -------- |
> |      | **Random**| **Sensitivity**     | **DPPN**     | **Random**| **Sensitivity** |  **DPPN**|
> |   0.1  | 0.2043 |   0.1689   |  **0.1270** | 0.5816  | 0.6511|  **0.6788**|
> |   0.2  | 0.2034 |  0.1532    |  **0.1165** | 0.5699  | 0.6558 |  **0.6705**|
> |   0.4  | 0.2091 | 0.1765     |  **0.1233** | 0.5424  | 0.6312 |  **0.6567**|
> |   0.6  | 0.2122 |  0.1732    |  **0.1297** | 0.5873  | 0.6156 |  **0.6462**|
> |   0.8  | 0.2127 |  0.1860    |  **0.1507** | 0.5849  | 0.6113 |  **0.6352**|
>
>
> **FIQA dataset**
> | **Ratio** | **Leakage $\downarrow$(%)** | ||  **Downstream $\uparrow$(%)** | ||
> | -------- | -------- | -------- | -------- | -------- |-------- | -------- |
> |      | **Random**| **Sensitivity**     | **DPPN**     | **Random**| **Sensitivity** |  **DPPN**|
> |   0.1 | 0.4285 | 0.2434     | **0.1783**  | 0.2208  | 0.2329|  **0.2583**|
> |   0.2 | 0.4364 | 0.2189     | **0.1893**  | 0.2233  | 0.2317 |  **0.2540**|
> |   0.4 | 0.4454  |  0.2276    | **0.1913**  | 0.2232  | 0.2289|  **0.2497**|
> |   0.6 | 0.4324 |  0.2613    | **0.2344**  | 0.2400  | 0.2385|  **0.2494**|
> |   0.8 | 0.4806 |   0.2975   | **0.2715**  | 0.2330  | **0.2453**|  0.2418|

---

> > ### Comment · Reviewer_FFaT · 2024-11-22
> >
> > Thank you for thoroughly addressing my concerns and providing detailed explanations. The additional experiments and analyses have significantly strengthened the manuscript, resolving my previously raised issues.
> >
> > In light of these improvements, I am increasing the rating to 6.

---

> > > ### Author Response · Authors · 2024-11-26
> > >
> > > Thank you very much for the score improvement and your constructive feedback. We will further polish the paper in the final revision. Thank you!

---

> ### Author Response · Authors · 2024-11-21
>
> **Q1: What if the privacy neurons were chosen based on a threshold of m, such as 0.5?**
>
> We appreciate the reviewer’s suggestion regarding threshold-based neuron selection. To address this, we conducted experiments with varying thresholds $h$, where the privacy neurons $\mathcal{N}_t$ are selected if the mask score $m_i$ for the $i$-th neuron is larger than $h$. The results, obtained using the STS12 and FIQA datasets under a perturbation level of  $\epsilon = 2 $, are summarized in the tables below. For comparison, we also include the performance of unprotected embeddings and the Laplace Mechanism (LapMech) baseline.
>
> **Results and Discussion.**
> From the results summarized below, we make the following observations:
> *  **Using a threshold-based neuron selection method proves to be an effective strategy.**  When setting the threshold $h \leq 0.6$, DPPN consistently outperforms the LapMech with lower privacy metrics (Leakage & Confidence) while maintaining higher downstream utility.
> *  **Optimal threshold selection.** We notice that most of the mask score $m_i$ for the $i$-th neuron falls into the range of 0.4~0.7. According to the experiment results setting the threshold to 0.5 or 0.6 seems to be a robust choice.
> We hope these results could address the reviewer’s concerns regarding threshold-based selection.
>
>
> **STS12 dataset:**
> | **Threshold** |  **Leakage $\downarrow$(%)**  | **Confidence $\downarrow$(%)** | **Downstream $\uparrow$(%)** |
> |----------|----------|----------|----------|
> | **unprotected**  | 60.09  | 47.81  | 74.25 |
> | **LapMech**  | 22.34  | 9.39  | 60.72 |
> | 0.9  | 59.83  | 47.54  | 74.25 |
> | 0.8  | 57.19  | 48.65  | 73.62 |
> | 0.7  | 43.89  | 12.78  | 70.51 |
> | 0.6  | 14.68  | 1.36  | 65.92 |
> | 0.5  |  12.33  | 0.08 | 63.17 |
> | 0.4  |  12.97 | 0.07 | 62.85 |
> | 0.3  |  13.48 | 1.11 | 61.43 |
> | 0.2  |  15.07 | 2.64 | 62.56 |
> | 0.1  |  17.45 | 2.88 | 61.49 |
>
>
>
> **FIQA dataset:**
> | **Threshold** |  **Leakage $\downarrow$(%)**  | **Confidence $\downarrow$(%)** | **Downstream $\uparrow$(%)** |
> |----------|----------|----------|----------|
> | **unprotected**  | 77.35  | 54.48  | 33.56 |
> | **LapMech**  | 35.17  | 16.70  | 21.74 |
> | 0.9  | 76.84  | 54.52  | 33.50|
> | 0.8  | 66.51 | 47.78  | 31.87 |
> | 0.7   | 31.79  | 19.10  | 28.65 |
> | 0.6   | 17.83  | 11.23  | 29.65 |
> | 0.5   | 18.93  | 11.24 | 27.74 |
> | 0.4   | 19.13 | 11.20 | 25.12 |
> | 0.3   | 23.44 | 10.85 | 25.49 |
> | 0.2   | 27.16 | 13.87 | 23.87 |
> | 0.1   | 28.78 | 14.33 | 22.51 |
>
>
>
> [1] Louizos, Christos, Max Welling, and Diederik P. Kingma. "Learning Sparse Neural Networks through L_0 Regularization." International Conference on Learning Representations. 2018.

---

### Official Review · Reviewer_8zq5 · 2024-10-23

**Soundness:** 3
**Presentation:** 2
**Contribution:** 3
**Rating:** 6
**Confidence:** 5

**Summary:**

The paper presents DPPN, a method for defending text embeddings against inversion attacks by selectively perturbing privacy-sensitive neurons. It demonstrates strengths in improving privacy-utility tradeoffs and shows robustness across different datasets and embedding models. However, it also has weaknesses, including the lack of theoretical privacy guarantees and potential challenges in generalizing to new data. Overall, DPPN is a promising approach for privacy protection in text embeddings, but further research is needed to address its limitations and ensure broader applicability.

**Strengths:**

Strengths:

1. Targeted Defense: DPPN focuses on identifying and perturbing only a subset of privacy-sensitive neurons, which is a more efficient approach compared to perturbing all dimensions of the embedding.

2. Improved Privacy-Utility Tradeoff: The paper demonstrates that DPPN achieves a superior balance between privacy protection and maintaining the utility of the embeddings for downstream tasks.

3. Effective Against Real-World Threats: DPPN shows significant effectiveness in mitigating sensitive information leakage on real-world datasets, such as medical records and financial data.

**Weaknesses:**

### 1. Personal Information Matching Issue

The DPPN method may not explicitly detail how to accurately match and process personal information. Ensuring accuracy and security, while protecting privacy, requires careful handling. This involves selecting and optimizing exact and fuzzy matching algorithms to provide accurate results without compromising privacy. The method uses a differentiable neuron mask learning framework to detect privacy neurons related to sensitive information. By assessing the importance of each embedding dimension, the top \( k \) dimensions with the highest importance are selected as privacy neurons. Visualizing the proportion of protected private data is necessary. The effectiveness of DPPN relies on accurately identifying and selecting neurons related to privacy information. Inaccuracies in this process may lead to sensitive information leakage.

### 2. Lack of Theoretical Guarantees

DPPN does not provide theoretical guarantees like Differential Privacy (DP), meaning it cannot quantify the degree of privacy protection or ensure the statistical insignificance of including or excluding a single data point. Adapting DPPN's targeted perturbation method to meet DP standards is challenging, as DP requires perturbing all published data for strong privacy guarantees, whereas DPPN only perturbs a subset of dimensions.

### 3. Challenges in Multilingual and Multicultural Backgrounds

Expressions of personal information vary across languages and cultural backgrounds. It is essential to discuss whether the DPPN method can adapt to and effectively handle these differences.

### 4. Real-Time Performance and Computational Cost

The real-time performance and computational cost of DPPN in practical applications are unclear. This is an important consideration for systems that need to process large volumes of data in real-time. The interpretability of the DPPN method is relatively low, potentially limiting its use in scenarios requiring high model interpretability, such as medical diagnosis.

### 5. Embedding Methods

The authors use embedding methods like GTR-base, Sentence-T5, and SBERT. It is suggested that traditional methods such as GloVe and Word2Vec be discussed and experimentally analyzed.

### 6. Presentation Issues

Table 9 is too large. Using \resizebox \textwidth is not recommended. Table 8: Statistics of datasets. The caption is below the table; the authors should unify the format. Authors can add more related work in the context:
- Private Language Models via Truncated Laplacian Mechanism EMNLP 2024
- Differentially Private Language Models Benefit from Public Pre-training PrivateNLP 2020

The explanations for Figures 1 and 3 are not detailed enough, especially regarding the meaning of arrows and labels. It is recommended that the authors clarify these in the legend.

**Questions:**

1. What are the benefits of neuron mask learning and direct matching for replacing PII compared to constructing a series of regular expressions to match and transform PII in the corpus?
2. What are the benefits or advantages of the author's method compared to traditional DP-based methods?
3. Could the author fully describe the challenges faced in this research?
4. The author would be greatly appreciated if the code could be made open source and contributed to the community.
5. Six datasets were used in the experiments, but only two datasets' results are shown in the main text. Could the author explain in more detail the reasons for choosing these two datasets?
6. The results in Table 1 show the performance of different methods, but there is a lack of in-depth analysis of these results. It is suggested to add a discussion of the results, explaining why DPPN performs better in these cases.

---

> ### Author Response · Authors · 2024-11-21
>
> We thank the reviewer for their feedback and will respond to the raised questions below.
>
> **Q1: What are the benefits of DPPN compared to matching and transforming PII in the corpus?**
>
> We acknowledge that matching and transforming PII is an effective approach to prevent privacy leakage, as it removes sensitive information entirely from the original text. However, this method has significant drawbacks, such as the loss of critical contextual information, which can substantially degrade the performance of downstream tasks. For instance, in medical retrieval tasks, removing details such as disease names or patient descriptions can severely impact the utility and effectiveness of the system.
>
> To empirically evaluate the impact of removing PII on downstream tasks, we implemented a text anonymizer based on the state-of-the-art tool provided by Azure Language Service [1], where all privacy tokens were replaced by the symbol ‘*’.
>
> We compare the privacy leakage and downstream performance of four methods: unprotected embeddings, embeddings with PII removed, and DPPN at two perturbation levels ($\epsilon=2$ and $\epsilon=1$). The results are summarized in the tables below, leading to the following key observations:
>
> 1. **Removing PII significantly impacts downstream utility.** Comparing the unprotected embeddings with the RemovePII method, we observe a substantial drop in downstream performance—from 74% to 59% on the STS12 dataset, and from 33% to 21% on the FIQA dataset. While RemovePII effectively mitigates privacy leakage, it suffers from severe information loss, resulting in low utility.
>
> 2. **DPPN provides an adaptive privacy-utility trade-off.** As shown in the tables, DPPN allows for balancing privacy and utility by adjusting the perturbation level $\epsilon$. For instance, with $\epsilon=2$, DPPN achieves better downstream performance than RemovePII while maintaining a relatively low privacy leakage (13.44% on the STS12 dataset). This adaptive mechanism enables data owners to fine-tune the balance between protecting privacy and maintaining utility according to their specific requirements.
>
> ### Results on STS12 Dataset
> | **Method**           | **Leakage $\downarrow$(%)** | **Downstream $\uparrow$(%)** |
> |-----------------------|-----------------|---------------------|
> | **Unprotected**       | 60.09           | 74.25              |
> | **RemovePII**         | -               | 59.47              |
> | **DPPN ($\epsilon=2$)** | 13.44           | 67.05              |
> | **DPPN ($\epsilon=1$)** | 1.61            | 40.78              |
>
> ### Results on FIQA Dataset
> | **Method**           | **Leakage $\downarrow$(%)** | **Downstream $\uparrow$(%)** |
> |-----------------------|-----------------|---------------------|
> | **Unprotected**       | 77.35           | 33.56              |
> | **RemovePII**         | -               | 21.24              |
> | **DPPN ($\epsilon=2$)** | 20.15           | 25.96              |
> | **DPPN ($\epsilon=1$)** | 2.01            | 15.05              |
>
>
> **Q2: What are the benefits or advantages of the author's method compared to traditional DP-based methods?**
>
> The advantages of our proposed DPPN over traditional DP-based methods are twofold:
>
> 1. **Enhanced Privacy-Utility Tradeoff:** Unlike traditional DP methods that perturb all dimensions indiscriminately, DPPN selectively perturbs only the most informative dimensions. This approach significantly reduces the risk of privacy leakage while preserving utility for downstream tasks. Table 1 and our experimental results demonstrate that DPPN consistently achieves a better balance between privacy and utility.
>
> 2. **Robustness Against Diverse Attack Models:** We evaluated DPPN against three strong attack models. The results show that DPPN provides better defense performance than other DP-based methods. This robustness validates its effectiveness in handling real-world scenarios with varied attack strategies.
>
>
> **Q3: What are the challenges faced in this research?**
>
> This research encounters two primary challenges. First, as noted in the general response, extending the current framework to satisfy formal differential privacy guarantees remains an ongoing effort. While we recognize that ensuring these guarantees is critical for developing a robust defense framework, we are still addressing the curse of dimensionality and working to derive a formal proof for our perturbation function. Second, learning a high-quality neuron detection method is inherently challenging due to the lack of explicit semantic meaning in embedding dimensions. Empirically, token information can only be inferred by observing changes in embeddings with and without specific tokens. Eventually, we found that adopting a neuron mask learning scheme proved to be an effective approach, and achieves performance comparable to white-box neuron detection scenarios.

---

> ### Author Response · Authors · 2024-11-21
>
> **Q5: Only two datasets' results are shown in the main text. Could the author explain in more detail the reasons for choosing these two datasets?**
>
> We selected the STS12 and FIQA datasets for inclusion in the main text for two primary reasons. First, these datasets are widely used benchmarks in prior work on embedding attacks and defenses [2,3]. This allows us to maintain consistency with established studies and provide a meaningful comparison. Second, these datasets represent distinct downstream tasks for a comprehensive evaluation of defense utility: the STS12 dataset assesses performance on the semantic text similarity task, while the FIQA dataset is used for retrieval tasks. Additionally, we emphasize that two other datasets, PII-Masking300K and MIMIC-III clinical notes, were employed to assess privacy risks in real-world threat scenarios. This ensures that our study not only aligns with prior research but also explores the implications of our methods in practical, high-risk applications.
>
> **Q6: Lack of in-depth analysis of the results in Table 1.**
>
> We appreciate the reviewer’s feedback regarding the analysis of results in Table 1. The primary goal of this experiment is to highlight the superior performance of DPPN in achieving an improved privacy-utility tradeoff compared to baseline methods. As demonstrated in Table 1, DPPN consistently outperforms the baselines by achieving lower privacy leakage and better downstream performance. This improvement stems from DPPN’s selective perturbation paradigm, which focuses on perturbing only the top 20% most informative neurons. In contrast, baseline methods perturb all dimensions uniformly, which compromises utility while providing a general defense.
> We believe this selective approach is a key contribution of our work. Furthermore, we validate the effectiveness of DPPN’s perturbation function in Section 5.2 and analyze the quality of neuron detection in Section 5.1 to support our findings.
>
> **W1.1: The DPPN method may not explicitly detail how to accurately match and process personal information.**
>
> We thank the reviewer for highlighting this point. To extract sensitive information, we utilize named entity recognition (NER) models to identify and extract named entities effectively. The detailed process for matching and processing personal information using the DPPN method is thoroughly described in Appendix C. We would be glad to address any further questions or provide additional clarifications if needed.
>
> **W1.2: The effectiveness of DPPN relies on accurately identifying and selecting neurons related to privacy information. Inaccuracies in this process may lead to sensitive information leakage.**
>
> We agree that the effectiveness of DPPN heavily depends on the accurate identification of privacy-related neurons. To address this, we conducted a thorough investigation into the quality of our neuron detection methods in Section 5.1. Figures 4 and 5 also demonstrate that DPPN achieves performance comparable to white-box defenses. Specifically, on the STS12 dataset, DPPN exhibits only a 3–6% absolute difference in privacy leakage metrics and less than a 5% relative difference in downstream task performance.  For a more comprehensive analysis, we encourage the reviewer to refer to Lines 358–369.
>
>
>
> **W2: Lack of Theoretical Guarantees.**
>
> We agree that it is important to develop a formal privacy guarantee for a defense method. As outlined in the general response, we propose a potential direction to extend DPPN with formal differential privacy definitions. While we acknowledge the technical challenges involved in formalizing such a theoretical framework, we believe that the key insights provided by our work—particularly in identifying privacy neurons—lay a strong foundation for future research to develop more robust defense strategies in this domain.

---

> ### Author Response · Authors · 2024-11-21
>
> **W4.1: The real-time performance and computational cost of DPPN in practical applications are unclear.**
>
> We appreciate the reviewer for pointing out this important concern. In fact, the proposed DPPN defense method is designed to be efficient in both complexity and scalability. Specifically, the data owner can precompute and store the privacy neurons for each token in advance. During inference, the system only needs to sample noise and add it to the corresponding precomputed privacy neurons, which is computationally lightweight. As a result, the computational complexity of DPPN is identical to all of the baseline methods.
>
> **W4.2: The interpretability of the DPPN method is relatively low, potentially limiting its use in scenarios requiring high model interpretability.**
>
> We believe this may be a misunderstanding. Interpretability is actually one of the core contributions of the DPPN method. Specifically, the detected privacy neurons offer a meaningful interpretation of the embedding dimensions. While individual embedding dimensions may not have explicit predefined meanings, our findings suggest that the embeddings themselves store token information in distinct, specific dimensions. As illustrated in Figure 6, we observe that semantically similar words tend to share the same privacy neurons,  which can be leveraged for implicit privacy protection. We hope this clarifies the reviewer’s concern.
>
> **References**
>
> [1]  https://learn.microsoft.com/en-us/azure/ai-services/language-service/overview
>
> [2] Morris, John, et al. "Text Embeddings Reveal (Almost) As Much As Text." Proceedings of the 2023 Conference on Empirical Methods in Natural Language Processing. 2023.
>
> [3] Kim, Donggyu, Garam Lee, and Sungwoo Oh. "Toward privacy-preserving text embedding similarity with homomorphic encryption." Proceedings of the Fourth Workshop on Financial Technology and Natural Language Processing (FinNLP). 2022.

---

> ### Author Response · Authors · 2024-11-26
>
> Dear Reviewer 8zq5,
>
> Thank you once again for your insightful comments and suggestions, which have been immensely helpful to us. We have posted responses to the proposed concerns.
>
> We understand this may be a particularly busy time, so we deeply appreciate any time you can spare to review our responses and let us know if they adequately address your concerns. If there are any additional comments, we will make every effort to address them promptly.
>
> Best regards,
> The Authors

---

> > ### Comment · Reviewer_8zq5 · 2024-11-26
> > **Comment**
> >
> > Thanks for authors' reply and clarification.
> >
> > **Reply to Q1**: Replacing PII does not necessarily mean directly removing personal information; it can involve some random replacement of PII. Deleting PII may not be a good baseline for comparison.
> >
> > **Reply to Q2**: The Laplacian mechanism is often less effective than the Gaussian mechanism because it is a pure-DP mechanism, which is too strict for NLP tasks. It is recommended that the authors include a comparison with the Gaussian mechanism.  Could the author clarify why LapMech and PurMech were chosen because there are many benchmarks available for comparison?
> >
> > **Reply to Q4**: If the authors could open-source their artifact and contribute it to the community, that would be ideal, such as https://anonymous.4open.science/, etc.
> >
> > **Reply to W1.1**: The NER method can only match a limited number of PII-related tokens. What happens to PII that is not matched? For example, in Figure 1, age 43 could also be considered a form of private data to some extent.

---

> > > ### Author Response · Authors · 2024-11-30
> > >
> > > **Q1: Replacing PII does not necessarily mean directly removing personal information; it can involve some random replacement of PII. Deleting PII may not be a good baseline for comparison.**
> > >
> > > Thank you for your comment. We agree that the concept of "replacing" PII encompasses a broader range of methods, such as randomization or substitution. To address this issue, we implement two additional PII transformation baseline methods.
> > >
> > > 1. **random word replacement**: Replace PII with a random word sampled in the corpus.
> > > 2. **semantic word replacement**: Replace PII with a semantic similar word under the same named entity category.
> > >
> > > We present the experimental results on the STS12 and FIQA dataset in the following tables. To alleviate the impact of randomness, we report the average downstream performance of 10 runs with different random seeds. We have the following observations:
> > >
> > > **PII transformation methods also suffer from various levels of information loss and reduced downstream utility.**
> > > Both the Semantic and Random Replacement methods result in a noticeable reduction in downstream performance. For instance, on the STS12 dataset, the Semantic Replacement method causes a drop in downstream from 74% to 64%, while on the FIQA dataset, the downstream drops from 33% to 18%. Similar trends are observed with Random Replacement. Notably, these transformation methods tend to yield worse performance on the FIQA dataset compared to directly removing PII. Since FIQA is an information retrieval task, we hypothesize that replacing PII disrupts the semantic coherence of the text, thereby degrading retrieval quality. In contrast, PII transformations have a less pronounced impact on the STS12 dataset, which is focused on text similarity.
> > >
> > > **Effectiveness of DPPN as a soft protection method.**
> > > While PII transformation methods are designed to protect private information, they often result in information loss and reduced task performance. In contrast, DPPN operates at the embedding level, perturbing the data in a more nuanced and "soft" manner that better preserves the semantic integrity of the text. Additionally, DPPN allows for a controlled tradeoff between privacy and utility by adjusting the perturbation level ($\epsilon$).
> > >
> > > **DPPN provides better privacy-utility tradeoff rate.**
> > > To evaluate the privacy-utility tradeoff of different methods, we calculate the tradeoff ratio $R = \frac{\Delta \text{Leakage}}{\Delta \text{Downstream}}$, where $\Delta \text{Leakage}$ represents the reduction in leakage (i.e., the difference in leakage between unprotected and protected data), and $\Delta \text{Downstream}$ is the corresponding reduction in downstream performance. As shown in the tables below, DPPN exhibits a superior tradeoff rate of 6.59 on the STS12 dataset and 7.52 on the FIQA dataset, outperforming the PII transformation methods.
> > >
> > > We hope these results help clarify the effectiveness of DPPN in offering a better balance between privacy protection and downstream utility, particularly when compared to traditional PII transformation methods.
> > >
> > >
> > > **Results on STS12 Dataset**
> > > | **Method**           | **Leakage $\downarrow$(%)** | **Downstream $\uparrow$(%)** | **Tradeoff Rate $R$ $\uparrow$** |
> > > |-----------------------|-----------------|---------------------|---------------------|
> > > | **Unprotected**       | 60.09           | 74.25              | -|
> > > | **RemovePII**         | -               | 59.47              | 4.12|
> > > | **Random-Replacement**         | -               | 60.50        | 4.42      |
> > > | **Semantic-Replacement**         | -               | 64.46    | 6.22          |
> > > | **DPPN ($\epsilon=2$)** | 13.44           | 67.05     | **6.59**         |
> > >
> > > **Results on FIQA Dataset**
> > > | **Method**           | **Leakage $\downarrow$(%)** | **Downstream $\uparrow$(%)** | **Tradeoff Rate $R$ $\uparrow$** |
> > > |-----------------------|-----------------|---------------------|---------------------|
> > > | **Unprotected**       | 77.35           | 33.56              |-|
> > > | **RemovePII**         | -               | 21.24              | 6.27 |
> > > | **Random-Replacement**         | -               | 19.20         | 5.38     |
> > > | **Semantic-Replacement**         | -               | 18.37          | 5.09    |
> > > | **DPPN ($\epsilon=2$)** | 20.15           | 25.96              |**7.52** |
> > >
> > >
> > > ---
> > >
> > > **Q4: If the authors could open-source their artifact and contribute it to the community, that would be ideal, such as https://anonymous.4open.science/, etc.**
> > >
> > > Thank you for your suggestion. We fully understand the importance of open-sourcing the artifact and contributing to the community. However, due to institutional policies and approval processes, we are unable to release the code publicly before acceptance. Once the paper is officially accepted, we will be able to make the code available to the community. We appreciate your understanding and look forward to making the artifact accessible to the research community.

---

> > > > ### Author Response · Authors · 2024-11-30
> > > >
> > > > **Q2.1.1: The Laplacian mechanism is often less effective than the Gaussian mechanism because it is a pure-DP mechanism, which is too strict for NLP tasks. It is recommended that the authors include a comparison with the Gaussian mechanism.**
> > > >
> > > > We sincerely appreciate the reviewer’s valuable comment regarding the comparison with the Gaussian mechanism. In response, we have implemented the Gaussian mechanism (GaussianMech) and compare its defense performance with our DPPN method. To ensure a fair comparison, we first sample Gaussian noise and then apply the neuron suppression perturbation as described in Eq. 6 for DPPN. For the Gaussian mechanism, we vary the privacy budget $\epsilon$ while fixing $\delta = 10^{-5}$, and evaluate both leakage and downstream task performance across two datasets: STS12 and FIQA.
> > > > Our experiments show that DPPN consistently outperforms the Gaussian mechanism in terms of both leakage reduction and downstream task performance. The difference is particularly noticeable at lower $\epsilon$ values, such as $\epsilon = 0.5$ and $\epsilon = 1$, where DPPN demonstrates significantly lower leakage and better downstream task accuracy. We hope the provided results clarify the effectiveness of DPPN in comparison to the Gaussian mechanism, and we hope this addresses the reviewer’s concern.
> > > >
> > > > **STS12 dataset**
> > > > | **Epsilon** | **Leakage $\downarrow$ (%)**    |      | **Downstream $\uparrow$(%)**  |    |
> > > > |---------|---------------------------|-------------|-------------|--------------|
> > > > |     | **GaussianMech**  | **DPPN**  | **GaussianMech**  | **DPPN**  |
> > > > | 0.5       | 15.08    | **6.54**     | 48.35     | **58.15** |
> > > > | 1       | 31.98  | **24.15**   | 70.03   | **72.25**   |
> > > > | 2       | 43.87  | **42.55**   | 73.68   | **73.95**   |
> > > > | 3       | **49.42**  | 49.46   | 74.04   | **74.12**   |
> > > >
> > > >
> > > > **FIQA dataset**
> > > > | **Epsilon** | **Leakage $\downarrow$ (%)**    |      | **Downstream $\uparrow$(%)**  |    |
> > > > |---------|---------------------------|-------------|-------------|--------------|
> > > > |  **Methods**      | **GaussianMech**  | **DPPN**  | **GaussianMech**  | **DPPN**  |
> > > > | 0.5       | 25.65    | **8.92**     | 14.51     | **22.07** |
> > > > | 1       | 47.90  | **37.58**   | 30.32   | **30.87**   |
> > > > | 2       | **64.43**  | 64.83   | 33.23   | **33.67**   |
> > > > | 3       | **69.74**  | 71.24   | 33.38   | **33.98**   |
> > > >
> > > > **Q2.1.2:Could the author clarify why LapMech and PurMech were chosen because there are many benchmarks available for comparison?**
> > > >
> > > > Thank you for raising this important question about the choice of baseline methods. As discussed in Section 8 of the paper, privacy-preserving text embeddings can be broadly classified into two categories: noise injection and adversarial training. Since DPPN relies on noise injection to achieve privacy preservation in text embeddings, it is most appropriate to compare it against other noise injection-based methods.
> > > >
> > > > While there is a large body of work on privacy-aware word embeddings with noise injection and differntial privacy, fewer studies focus on sentence-level embeddings. The two baseline methods we selected—Laplace Mechanism (WSDM 2020) [1] and Purkayastha Mechanism (WWW 2023) [2]—are currently the most well-established and theoretically grounded approaches for this specific problem. This is further supported by Table 1 in the survey paper [3], which identifies [2] and [4] as the only perturbation-based methods for sentence-level privacy, with [1] being a general DP mechanism. We excluded [4] from our experiments due to its additional requirements, such as fine-tuning the embedding model and reliance on an external corpus.  Given the limited prior work in sentence-level privacy-preserving embeddings, we believe these selected baselines represent the most relevant and comparable methods for our study.
> > > >
> > > > **References**
> > > >
> > > > [1] Feyisetan, Oluwaseyi, et al. "Privacy-and utility-preserving textual analysis via calibrated multivariate perturbations." Proceedings of the 13th international conference on web search and data mining. 2020.
> > > >
> > > > [2] Du, Minxin, et al. "Sanitizing sentence embeddings (and labels) for local differential privacy." Proceedings of the ACM Web Conference 2023. 2023.
> > > >
> > > > [3] Hu, Lijie, et al. "Differentially Private Natural Language Models: Recent Advances and Future Directions." Findings of the Association for Computational Linguistics: EACL 2024. 2024.
> > > >
> > > > [4]  Meehan, Casey, Khalil Mrini, and Kamalika Chaudhuri. "Sentence-level Privacy for Document Embeddings." Proceedings of the 60th Annual Meeting of the Association for Computational Linguistics (Volume 1: Long Papers). 2022.

---

> > > > > ### Author Response · Authors · 2024-11-30
> > > > >
> > > > > **W1.1: The NER method can only match a limited number of PII-related tokens. What happens to PII that is not matched? For example, in Figure 1, age 43 could also be considered a form of private data to some extent.**
> > > > >
> > > > > We thank the reviewer for pointing out an important scenario where the sensitive information is difficult to extract and sometimes difficult to match. We would like to address two important points.
> > > > >
> > > > > First, we acknowledge that our experiments assume the data owner can extract sensitive information using rule-based systems or NER frameworks. Under this assumption, our results show that DPPN is effective in preventing privacy leakage for various categories of private information, including age, sex, and disease, as detailed in Table 4 and Section 6.2.
> > > > >
> > > > > Second, we agree that there is a potential risk where PII may not be fully matched by the system. However, we would like to highlight an important finding in this work: **semantic
> > > > > similar words share similar privacy neuron**. As detailed in Section 5.3 and Figure 6, we observed that these **"privacy neurons" represent specific privacy concepts**, meaning that perturbing a privacy neuron for one word can also provide protection for other semantically related terms. For example, the privacy neuron associated with "america" can also help protect related terms like "U.S." and "American" (see Figure 6).
> > > > >
> > > > > Additionally, we performed an experiment to verify the implicit protective capability of our approach, which is shown in Table 9. The results demonstrate that **when DPPN suppresses privacy neurons for a specific term, it also extends protection to semantically related words**, even if they are not explicitly matched. We hope this explanation clarifies how DPPN can provide broader privacy protection, even in cases where PII is not perfectly matched, and addresses the reviewer’s concern.

---

> > > > > > ### Comment · Reviewer_8zq5 · 2024-12-03
> > > > > >
> > > > > > Thanks for the authors' feedback. The authors have addressed some of my concerns. I will keep my score.

---

### Official Review · Reviewer_QCCL · 2024-10-24

**Soundness:** 2
**Presentation:** 2
**Contribution:** 2
**Rating:** 6
**Confidence:** 3

**Summary:**

This paper introduces DPPN (Defense through Perturbing Privacy Neurons), a novel method that protects text embeddings by selectively identifying and perturbing privacy-sensitive neurons. The experiments show the effectiveness of the method.

**Strengths:**

1. This paper has a clear format, highlighting important keywords throughout the paper.
2. The paper explains the methodology in details.

**Weaknesses:**

1. Is LapMech a variant of DP? What is the performance comparison with standard DP? The paper can add a new section on this.
2. What is "Downstream" in utility metric? The paper could explain more about how they measure the utility of the method.
3. How do you explain the results in Table 7? Table 7 gives more results of the method than Table 1, and It seems that DPPN does not outperform other defense methods in some datasets.
4. The paper can include some other defense methods in their comparison. Experiments with two baselines are not very convincing.
5. In Sec 4.1, they mention that "Vec2text serves as our primary attack model in subsequent experiments." So it seems that only one attack model is evaluated in the whole paper, making the defense performance not convincing again. Are there any results for other attack models?
6. Only Table 3 shows some results for attack models. However, only privacy metrics are presented. How about the utility metrics?
7. The paper could reorganize some experimental results based on some comments above, making the experiment section better.

**Questions:**

Please see those in weaknesses.

---

> ### Author Response · Authors · 2024-11-21
>
> We thank the reviewer for their feedback and will respond to the raised questions below.
>
> **W1: Is LapMech a variant of DP? What is the performance comparison with standard DP?**
>
> LapMech is indeed based on the Laplace mechanism, which is one of the foundational approaches in differential privacy (DP) and provides pure $\epsilon$-DP guarantees.  Both baseline methods in our evaluation, LapMech and PurMech, are DP-based approaches tailored for privacy preservation in text embeddings. Among them, PurMech represents the state-of-the-art defense against inversion attacks on embeddings.
>
> Regarding performance comparisons with standard DP mechanisms, we believe that the evaluations presented in Table 1 and Table 7 of our manuscript offer a comprehensive analysis. These results demonstrate that DPPN significantly outperforms the DP-based baselines in both privacy and utility metrics.
>
> **W2: What is "Downstream" in the utility metric?**
>
> Downstream refers to the subsequent tasks that utilize the text embeddings as input features. Since text embeddings serve as general-purpose representations, their utility is often evaluated through their performance on various downstream applications, including text classification, information retrieval, and semantic similarity assessment. Details of the downstream tasks and their evaluation metrics for each dataset could be found in Table 8 of our manuscript. In our evaluation, we specifically follow the comprehensive evaluation protocol established by the Massive Text Embedding Benchmark (MTEB) [1]. This benchmark provides standardized metrics for assessing embedding quality across multiple downstream tasks, allowing for systematic comparison of embedding utility.
>
> **W3:  It seems that DPPN does not outperform other defense methods in some datasets. How do you explain the results in Table 7?**
>
> We acknowledge that DPPN does not consistently outperform all baseline methods across every dataset and perturbation level. However, it is important to emphasize that DPPN demonstrates superior performance in the majority of cases. Specifically, in Table 7, DPPN outperforms the baseline methods in 48 out of 60 cases, achieving an **80% win rate**.
>
> For the minority of cases where DPPN underperforms, the relative difference compared to the best-performing baseline is only **8.05%**. In contrast, when DPPN outperforms the second-best baseline, it achieves a substantial gains with **average improvement of 31.21%** in these scenarios. We also observe that DPPN’s underperformance typically occurs at lower perturbation levels (e.g., $\epsilon = 6$ or $\epsilon = 8$). Since the injected noise is minimal, the advantage of DPPN is reduced. This limitation is common to noise-based privacy schemes, where the trade-off between privacy and utility diminishes with lower noise levels.
>
> In summary, while DPPN may not always achieve the top performance in every setting, we believe its overall performance and significant improvements in the majority of cases could validate its effectiveness and robustness compared to baseline methods.
>
> [1] Muennighoff, Niklas, et al. "MTEB: Massive Text Embedding Benchmark." Proceedings of the 17th Conference of the European Chapter of the Association for Computational Linguistics. 2023.

---

> ### Author Response · Authors · 2024-11-21
>
> **W4: Experiments with two baselines are not very convincing.**
>
> We thank the reviewer for raising this concern regarding the selection of baseline methods. As discussed in the related works section (Section 8), there are two main approaches for privacy-preserving text embeddings: noise injection and adversarial training. Since DPPN achieves privacy preservation by selectively injecting noise into text embeddings, it is most appropriate to compare DPPN against noise injection-based methods.
>
> While there is an extensive body of research on privacy-aware word embeddings, there are significantly fewer works focused on sentence-level embeddings. As a result, the two baseline methods used in our experiments—**Laplace Mechanism** (WSDM 2020) [2] and **Purkayastha Mechanism** (WWW’23) [3]—are currently the most theoretically formalized and publicly recognized methods for this problem. This is further supported by Table 1 in the survey paper [4], which identifies [3] and [5] as the only two perturbation-based methods for sentence-level privacy, with [2] being a general DP mechanism. It is worth noting that we excluded [5] from our experiments due to its additional requirements, including fine-tuning the embedding model and reliance on an external corpus. Given the scarcity of prior research in sentence-level privacy-preserving text embeddings, our choice of baselines reflects the state-of-the-art approaches available.
>
> Experiment results with additional baseline. To address the concern and further enhance our evaluation, we implemented an additional baseline method that perturbs text embeddings by adding Gaussian noise. This method was studied in the Vec2Text [6] paper as a promising privacy defense mechanism. However, it is important to note that this method does not offer formal guarantees of differential privacy (DP). The results of this additional baseline are presented in the tables below.
> Our analysis shows that the NormalNoise baseline exhibits similar performance to other baseline methods. Notably, our proposed DPPN consistently outperforms NormalNoise across all evaluation metrics. These results reinforce the robustness of DPPN and further validate its effectiveness compared to other noise injection-based approaches.
> We hope this addresses the reviewer’s concerns and demonstrates our rationale for baseline selection.
>
> **STS12 dataset**
> | Epsilon | **Leakage $\downarrow$(%)**    |      |     |     | **Confidence $\downarrow$(%)**   |     |      |     | **Downstream $\uparrow$(%)**  |    |     |     |
> |---------|---------------------------|-------------|-------------|--------------|---------------------|-------------|-------------|-------------|---------------------|-------------|-------------|-------------|
> |        | **NormalNoise**  | **LapMech**  | **PurMech**  | **DPPN**  | **NormalNoise**  | **LapMech**  | **PurMech**  | **DPPN**  | **NormalNoise**  | **LapMech**  | **PurMech**  | **DPPN**  |
> | 1       | 7.52    | 7.36     | 7.42     | **1.61**     | 6.67   | 6.70     | 6.80    | **6.05**    | 29.24  | 29.28  | 29.31  | **40.78**  |
> | 2       | 22.83  | 22.34   | 22.66   | **13.44**   | 9.13   | 9.39     | 9.42    | **8.25**    | 60.65  | 60.72  | 60.72  | **67.05**  |
> | 4       | 38.89  | 38.17   | 38.04   | **33.49**   | 24.87  | 24.70   | 24.74  | **23.80**  | 72.45  | 72.47  | 72.47  | **73.40**  |
> | 6       | 45.39  | 44.74   | 44.76   | **42.59**   | 33.79  | 34.59   | 34.59  | **34.14**  | 73.67  | 73.68  | 73.68  | **73.95**  |
> | 8       | 49.29  | 48.48| 48.48  | **48.34**   | 47.11   | 39.08  | 38.75    | **38.49** | 38.49   | 73.97  | **73.98** | **73.98**  | 74.09  |
>
> **FIQA dataset**
> | Epsilon | **Leakage $\downarrow$(%)**    |      |     |     | **Confidence $\downarrow$(%)**   |     |      |     | **Downstream $\uparrow$(%)**  |    |     |     |
> |---------|---------------------------|-------------|-------------|--------------|---------------------|-------------|-------------|-------------|---------------------|-------------|-------------|-------------|
> |        | **NormalNoise**  | **LapMech**  | **PurMech**  | **DPPN**  | **NormalNoise**  | **LapMech**  | **PurMech**  | **DPPN**  | **NormalNoise**  | **LapMech**  | **PurMech**  | **DPPN**  |
> | 1       | 13.33   | 12.56   | 13.01   | **2.01**   | 6.48    | 6.67    | 6.70     | **5.84**    | 10.74  | 10.64  | 10.63    | **15.05**    |
> | 2       | 34.58   | 35.17   | 35.31   | **20.15**  | 16.96   | 16.70   | 16.55    | **11.92**   | 21.47  | 21.74  | 21.76    | **25.96**    |
> | 4       | 55.11   | 55.69   | 55.38   | **51.26**  | 35.20   | 35.32   | 35.25    | **31.36**   | 32.10  | 32.22  | 32.23    | **32.84**   |
> | 6       | 64.53   | 64.12   | 64.13   | **62.79**  | 44.23   | 43.35   | 43.56    | **41.57**   | 33.10  | 33.24  | 33.26    | **33.58**    |
> | 8       | 68.74   | 68.85   | 68.63   | **67.99**  | 48.22   | 48.07   | 47.77    | **46.25**   | 33.34  | 33.50  | 33.52    | **33.73**    |

---

> ### Author Response · Authors · 2024-11-21
>
> **W5: It seems that only one attack model is evaluated in the whole paper, making the defense performance not convincing again. Are there any results for other attack models?**
>
> We believe there is a misunderstanding regarding the evaluation of attack models in our paper. As clarified in **Table 3** and **Section 6.1**, our manuscript evaluates **three attack models**: Vec2text [6], GEIA [7], and MLC [8]. Due to page constraints, we only presented experimental results on the STS12 dataset with $\epsilon = 1$ and $\epsilon = 2$ in the main text. To address the reviewer’s concern, we include additional experimental results using the GEIA and MLC attack models below. For brevity, we report only privacy-related metrics (Leakage & Confidence), as the utility metric remains consistent with those in Table 1 where DPPN outperforms the baseline methods.
>
> The results in the tables below demonstrate that DPPN consistently outperforms baseline methods (LapMech and PurMech) across various perturbation levels ($\epsilon$) and attack models. We hope this provides sufficient evidence to validate the effectiveness of DPPN and addresses the reviewer’s concern.
>
>
> **Results using MLC[8] as attack model:**
>
> **STS12 dataset**
> | Epsilon | **Leakage $\downarrow$(%)**                     |                 |                 |
> |---------|---------------------------------|-----------------|-----------------|
> |  **Methods**   | **LapMech**         | **PurMech**      | **DPPN**           |
> | 1       | 49.39     | 49.80  | **47.63**    |
> | 2       | 52.74     | 52.68  | **49.59**    |
> | 4       | 52.35     | 52.48 | **50.25**    |
> | 6       | 52.16     | 52.15 | **51.01**    |
> | 8       | 52.38     | 52.69 | **51.69**    |
>
> **FIQA dataset**
> | Epsilon | **Leakage $\downarrow$(%)**                     |                 |                 |
> |---------|---------------------------------|-----------------|-----------------|
> | **Methods**      | **LapMech**         | **PurMech**      | **DPPN**           |
> | 1       | **41.24**    | 41.94  | 45.51     |
> | 2       | 49.46   | 49.75  | **45.74**    |
> | 4       | 53.37   | 53.33  | **50.87**    |
> | 6       | 54.17    | 54.19  | **53.05**    |
> | 8       | 54.48    | 54.32 2 | **53.95**    |
>
>
>
>
> **Results using GEIA[7] as attack model:**
>
>
> **STS12 dataset**
> | Epsilon | **Leakage $\downarrow$(%)**                     |                 |                 |
> |---------|---------------------------------|-----------------|-----------------|
> | **Methods**   | **LapMech**         | **PurMech**      | **DPPN**           |
> | 1 | 12.30  | 12.36  | **7.08** |
> | 2 | 20.60  | 21.21  | **15.82** |
> | 4 | 23.61  | 23.51  | **21.49** |
> | 6 | 24.77  | 24.84 | **22.97** |
> | 8 | 24.81  | 24.85  | **23.50** |
>
> **FIQA dataset**
> | Epsilon | **Leakage $\downarrow$(%)**                     |                 |                 |
> |---------|---------------------------------|-----------------|-----------------|
> |   **Methods**  | **LapMech**         | **PurMech**      | **DPPN**           |
> | 1 | 24.18  | 23.58  | **4.27** |
> | 2 | 41.27  | 40.85  | **25.13** |
> | 4 | 46.06  | 46.51  | **40.09** |
> | 6 | 47.43  | 47.31  |**43.27** |
> | 8 | 47.76  | 47.87 | **44.73** |
>
> **W6: Only Table 3 shows the privacy metrics results for different attack models.  How about the utility metrics?**
>
> The experiment presented in Table 3 specifically focuses on evaluating privacy vulnerabilities under different attack models. The utility metrics are not included in Table 3 because they remain constant across all attack models for a given defense method. These utility results are identical to those reported in Table 1 for the STS12 dataset, as the defense mechanism's impact on utility is independent of the attack model being evaluated.
>
> **References**
>
> [1] Muennighoff, Niklas, et al. "MTEB: Massive Text Embedding Benchmark."
>
> [2] Feyisetan, Oluwaseyi, et al. "Privacy-and utility-preserving textual analysis via calibrated multivariate perturbations."
>
> [3] Du, Minxin, et al. "Sanitizing sentence embeddings (and labels) for local differential privacy."
>
> [4] Hu, Lijie, et al. "Differentially Private Natural Language Models: Recent Advances and Future Directions."
>
> [5]  Meehan, Casey, Khalil Mrini, and Kamalika Chaudhuri. "Sentence-level Privacy for Document Embeddings."
>
> [6] Morris, John, et al. "Text Embeddings Reveal (Almost) As Much As Text."
>
> [7] Li, Haoran, Mingshi Xu, and Yangqiu Song. "Sentence Embedding Leaks More Information than You Expect: Generative Embedding Inversion Attack to Recover the Whole Sentence."
>
> [8] Song, Congzheng, and Ananth Raghunathan. "Information leakage in embedding models."

---

> ### Author Response · Authors · 2024-11-26
>
> Dear Reviewer QCCL,
>
> Thank you once again for your insightful comments and suggestions, which have been immensely helpful to us. We have posted responses to the proposed concerns.
>
> We understand this may be a particularly busy time, so we deeply appreciate any time you can spare to review our responses and let us know if they adequately address your concerns. If there are any additional comments, we will make every effort to address them promptly.
>
> Best regards,
> The Authors

---

> > ### Comment · Reviewer_QCCL · 2024-11-30
> >
> > Thanks for the authors' feedback. The authors have addressed some of my concerns. I have increased the score.

---

> > > ### Author Response · Authors · 2024-12-02
> > >
> > > Thank you for taking the time to review our work and for increasing the score. We sincerely appreciate your thoughtful consideration and the constructive feedback that has helped improve our manuscript. If there are any additional concerns or suggestions that could further enhance the quality of our work, we would be happy to address them.

---

### Official Review · Reviewer_GaoG · 2024-11-05

**Soundness:** 3
**Presentation:** 4
**Contribution:** 2
**Rating:** 6
**Confidence:** 3

**Summary:**

This paper focuses on defense strategies against embedding inversion attacks, a type of privacy attack where attackers attempt to reconstruct original sensitive data from its embedding representation. Existing defense methods commonly add noise uniformly across all embedding features. However, this approach is limited in maintaining model performance and is limited in effectively protecting privacy since, ideally, more noise should be directed towards privacy-sensitive features.

To address these issues, the authors first assume and validate that embeddings are composed of both privacy-sensitive and privacy-invariant features. Then, they propose an optimization problem in which a differentiable mask is optimized to isolate privacy-sensitive information within learned embeddings. The optimized mask becomes a tool to detect privacy-sensitive features, and by adding noise to these features, the authors achieve defense against embedding inversion attacks.

**Strengths:**

1. The method is technically sound, successfully enhances benign accuracy in downstream tasks, and prevents privacy leakage.

2. The hypothesis validated in this paper, that features can be divided into privacy-sensitive and privacy-invariant categories, is quite interesting. Building on this, the idea of using a mask to separate these features is also very novel.

**Weaknesses:**

1. The paper lacks a formal privacy guarantee. For instance, in methods like LapMech, the authors provide a proof to demonstrate the effectiveness of privacy protection, but this paper lacks such a discussion. Such a drawback raises doubts about the trustworthiness of the proposed method, especially if attackers know the defense mechanism.

2. This defense method requires two datasets, one containing privacy-sensitive data and one without, which necessitates labeling what information as private information. This adds a labeling burden in real-world datasets, as additional annotation is required.

**Questions:**

1. Is it possible to formulate privacy realized by this framework?

2. It is recommended that the authors consider adaptive attack scenarios to demonstrate that this defense method remains effective even if the privacy protection scheme is exposed to adversaries.

---

> ### Author Response · Authors · 2024-11-21
>
> We thank the reviewer for their feedback and will respond to the raised questions below.
>
> **W1 & Q1: The paper lacks a formal privacy guarantee. Is it possible to formulate privacy realized by this framework?**
>
> We agree that it is important to develop a formal privacy guarantee for a defense method. As outlined in the general response, we propose a potential direction to extend DPPN with formal differential privacy definitions. While we acknowledge the technical challenges involved in formalizing such a theoretical framework, we believe that the key insights provided by our work—particularly in identifying privacy neurons—lay a strong foundation for future research to develop more robust defense strategies in this domain. Additionally, we conducted an experiment (addressed in the following question) to demonstrate that our method remains effective even when the defense mechanism is disclosed to adversaries. We hope this clarifies and addresses the reviewer’s concern.
>
> **W2: This defense method requires two datasets, which necessitates labeling what information as private information.**
> The proposed defense method involves two datasets to evaluate the embedding differences with and without private information. However, we argue that this does not impose an additional labeling burden, as the datasets can be constructed using publicly available or generated resources.
>
> For instance, if the private information to be protected pertains to medical data, the data owner can utilize publicly available medical-related articles, such as those on Wikipedia, or curated medical datasets. Additionally, synthetic privacy-related texts can be generated using generative models like ChatGPT, which can simulate relevant private information. This approach enables the data owner to construct the positive dataset  $D^+$. The negative dataset $ D^-$ can then be derived from $D^+$  through simple modifications, as outlined in Lines 205–206 of the manuscript. In summary, the requirement of two datasets does not require additional manual labeling.

---

> ### Author Response · Authors · 2024-11-21
>
> **Q2:  Is it possible to design an adaptive attack scenario to demonstrate that this defense method remains effective even if the privacy protection scheme is exposed to adversaries.**
>
> We appreciate the reviewer highlighting this scenario to evaluate DPPN’s robustness when adversaries are aware of the protection scheme. To address this, we consider an adaptive attack where the adversary knows which embedding dimensions (i.e., the privacy neurons $\mathcal{N}_t$) are perturbed.
>
> The research question is: **Can the adversary extract private information from the “unperturbed” embeddings?** As defined in Definition 1, text embeddings decompose into privacy-sensitive embeddings and privacy-invariant embeddings. Since the privacy-sensitive embeddings are perturbed through noise injection, the adversary would resort to train their attack model on privacy-invariant embeddings.
>
> **Adaptive attack experiment setup:**
>
> The attacker is given the full text embedding and access to the indices of the top-10% privacy neurons. The attacker builds a threat model by excluding these privacy neurons from their training (i.e., training is performed on $d*0.9$ dimensions). We term the scenario as oracle since the attacker has additional knowledge on the defense mechanism. We also report the attack performance when the defense mechanism is unknown to the adversary.
>
> **Results and discussion:**
>
> * **With external knowledge on the defense mechanism, the adversary did attack better compared to the unknown scenario.** As expected, an adversary with external knowledge of the defense mechanism performs better than one without it. On the STS12 dataset, privacy leakage increased to 25% in the oracle scenario, compared to 13% in the unknown scenario. A similar trend is observed in the FIQA dataset.
> * **The attacker cannot inference private information from privacy-invariant embeddings.** Despite oracle knowledge, the adversary’s ability to infer private information is significantly reduced compared to unprotected embeddings. For example, in the STS12 dataset, leakage drops from 60% (unprotected) to 25% (oracle), demonstrating that excluding the privacy neurons significantly limits the adversary’s success. A similar reduction is observed in the FIQA dataset, where leakage drops from 77% to 28%. These results indicate that even with full knowledge of the protection scheme, the adversary cannot reconstruct private information from the privacy-invariant embeddings.
>
>
> **STS12 dataset:**
>
> | Method |  Leakage$\downarrow$ (%)  | Confidence$\downarrow$ (%) | Downstream$\uparrow$ (%) |
> |----------|----------|----------|----------|
> | unprotected  | 60.09  | 47.81  | 74.25 |
> | oracle-attack   | 25.03  | 16.37  | 73.51 |
> | unknown-attack  | 13.44 | 6.05 | 67.05 |
>
> **FIQA dataset:**
> | Method |  Leakage$\downarrow$ (%) | Confidence$\downarrow$ (%) | Downstream$\uparrow$ (%) |
> |----------|----------|----------|----------|
> | unprotected  | 77.35  | 54.48  | 33.56 |
> | oracle-attack   | 28.86 | 19.12 | 29.64 |
> | unknown-attack  | 20.15 | 11.92 | 25.96 |

---

> ### Author Response · Authors · 2024-11-26
>
> Dear Reviewer GaoG,
>
> Thank you once again for your insightful comments and suggestions, which have been immensely helpful to us. We have posted responses to the proposed concerns.
>
> We understand this may be a particularly busy time, so we deeply appreciate any time you can spare to review our responses and let us know if they adequately address your concerns. If there are any additional comments, we will make every effort to address them promptly.
>
> Best regards,
> The Authors

---

> > ### Comment · Reviewer_GaoG · 2024-11-27
> >
> > The authors have addressed some of my concerns. Although their response regarding the theoretical part was not entirely satisfactory, their experiments on adaptive attacks are convincing. Therefore, I have raised my score to 6.

---

> > > ### Author Response · Authors · 2024-11-28
> > >
> > > Thank you for your thoughtful feedback.  We are grateful for your recognition of the strength of our experiments on adaptive attacks and the updated score. While we understand that the theoretical aspect may not have fully addressed your concerns, we are glad that the experimental results have helped to strengthen the overall contribution of the paper.

---

### Author Response · Authors · 2024-11-21
**General Response on the theoretical guarantees of DPPN**

We acknowledge that the absence of a formal theoretical privacy guarantee represents a limitation of our work. Below, we outline a promising direction for incorporating formal privacy guarantees into DPPN and discuss the associated challenges in achieving this goal.


**Extending DPPN with Regularized Mahalanobis Differential Privacy:**

To meet standard differential privacy requirements, all dimensions of the text embeddings must be perturbed rather than focusing on a subset of embedding dimensions. A potential approach involves introducing "elliptical" noise into the text embeddings. Unlike the traditional Laplace mechanism, which adds isotropic (spherical) noise sampled uniformly from a multivariate Laplace distribution, elliptical noise allows for dimension-wise perturbation with variable scales. The Mahalanobis distance allows for more flexible, dimension-wise perturbations by accounting for the covariance structure of data, rather than treating all dimensions equally. This technique has been studied in the context of word embeddings [1], where privacy is defined under a metric-local differential privacy (metric-LDP) framework using the Mahalanobis distance (Definitions 3 and 4 of [1]). While this direction is promising, several challenges must be addressed.
Technical Challenges:
1) **Curse of Dimensionality in High-Dimensional Data:**  Differential privacy in high-dimensional spaces faces significant challenges, as the noise magnitude required for  privacy increases linearly with the embedding dimension [2]. Specifically, the noise norm in the Laplace mechanism follows the Gamma distribution $\Gamma(d, \epsilon)$, where $d$ is the embedding dimension and $\epsilon$ is the privacy budget.
This requires setting a very low privacy budget (e.g., $\epsilon \geq 500$) to maintain embedding utility for modern text embedding models with large dimensions (e.g., $d=768$ for SentenceBERT). While dimension reduction methods like Gaussian random projection could mitigate this issue, they are incompatible with DPPN's core design because they disrupt the separation of privacy-sensitive and privacy-invariant dimensions that is central to DPPN.

2) **Deriving a Formal Proof for the Neuron-Suppressing Perturbation Function:**
As demonstrated in Sections 3.3 and 5.2, our perturbation function achieves effective defense against inversion attacks compared to existing DP-based mechanisms. However, constructing a formal proof that this approach satisfies the Regularized Mahalanobis differential privacy guarantee is non-trivial. Specifically, this would require defining the covariance matrix for the Mahalanobis distance and formulating a corresponding perturbation function that meets metric-LDP criteria. We acknowledge this as a critical avenue for future work and intend to explore formalizing these guarantees in subsequent research.

**Discussion:**

We believe the core contribution of our work—defending against inversion attacks through dimension-selective perturbation—addresses a critical gap in the literature. Moreover, it is worth noting that other prominent works, such as [3, 4, 5], have also made meaningful contributions to privacy research despite the lack of formal guarantees. We hope that our findings will inspire further exploration and innovation in embedding-based privacy defense mechanisms.

**References**

[1] Xu, Zekun, et al. "A Differentially Private Text Perturbation Method Using Regularized Mahalanobis Metric." Proceedings of the Second Workshop on Privacy in NLP. 2020.

[2] Wu, Xi, et al. "Bolt-on differential privacy for scalable stochastic gradient descent-based analytics." Proceedings of the 2017 ACM International Conference on Management of Data. 2017.

[3] Coavoux, Maximin, Shashi Narayan, and Shay B. Cohen. "Privacy-preserving Neural Representations of Text." Proceedings of the 2018 Conference on Empirical Methods in Natural Language Processing. 2018.

[4] Elazar, Yanai, and Yoav Goldberg. "Adversarial Removal of Demographic Attributes from Text Data." Proceedings of the 2018 Conference on Empirical Methods in Natural Language Processing. 2018.

[5] Struppek, Lukas, Dominik Hintersdorf, and Kristian Kersting. "Be Careful What You Smooth For: Label Smoothing Can Be a Privacy Shield but Also a Catalyst for Model Inversion Attacks." The Twelfth International Conference on Learning Representations.

---

### Meta-Review · Area_Chair_itwg · 2024-12-20

**Metareview:**

This is a borderline paper. While the reviewers agreed on the merits of the problem, and some of the algorithmic approaches, there were legitimate concerns about the lack of a formal privacy guarantee. Since it seems methods developed in the differential privacy literature is applicable, so a question that is glaring is how the current method compare to formal privacy preserving methods. The authors did summarize it as a future direction. It seems that having the formal privacy component is crucial to the paper.

**Additional Comments On Reviewer Discussion:**

NA

---

### Decision · Program_Chairs · 2025-01-22

Reject